**Letter**

# Exome sequencing identifies breast cancer susceptibility genes and defines the contribution of coding variants to breast cancer risk

**A list of authors and their affiliations appears at the end of the paper**

Linkage and candidate gene studies have identified several breast cancer susceptibility genes, but the overall contribution of coding variation to breast cancer is unclear. To evaluate the role of rare coding variants more comprehensively, we performed a meta-analysis across three large whole-exome sequencing datasets, containing 26,368 female cases and 217,673 female controls. Burden tests were performed for protein-truncating and rare missense variants in 15,616 and 18,601 genes, respectively. Associations between protein-truncating variants and breast cancer were identified for the following six genes at exome-wide significance ($P < 2.5 \times 10^{-6}$): the five known susceptibility genes *ATM*, *BRCA1*, *BRCA2*, *CHEK2* and *PALB2*, together with *MAP3K1*. Associations were also observed for *LZTR1*, *ATRIP* and *BARD1* with $P < 1 \times 10^{-4}$. Associations between predicted deleterious rare missense or protein-truncating variants and breast cancer were additionally identified for *CDKN2A* at exome-wide significance. The overall contribution of coding variants in genes beyond the previously known genes is estimated to be small.

Breast cancer is the leading cause of cancer-related mortality for women worldwide. Genetic susceptibility to breast cancer is known to be conferred by common variants, identified through genome-wide association studies (GWAS), together with rarer coding variants conferring higher disease risks. The latter, identified through genetic linkage or targeted sequencing studies, includes protein-truncating variants (PTVs) and some rare missense variants in *ATM*, *BARD1*, *BRCA1*, *BRCA2*, *CHEK2*, *RAD51C*, *RAD51D*, *PALB2* and *TP53* (ref. 1). However, these variants together explain less than half the familial relative risk (FRR) of breast cancer[2]. The contribution of rare coding variants in other genes remains largely unknown.

Here we used data from the following three large whole-exome sequencing (WES) studies, primarily of European ancestry, to assess the role of rare variants in all coding genes: the Breast Cancer Risk after Diagnostic Gene Sequencing (BRIDGES) dataset that included samples from eight studies in the Breast Cancer Association Consortium (BCAC), the PERSPECTIVE (Personalized Risk assessment for prevention and early detection of breast cancer: integration and implementation) dataset that included three BCAC studies (Supplementary Table 1) and UK Biobank (UKB). After quality control (QC; Methods), these datasets comprised 26,368 female cases and 217,673 female controls (Supplementary Table 2).

We considered the following two main categories of variants: PTVs and rare missense variants (minor allele frequency <0.001). Single-variant association tests are generally underpowered for rare variants; however, burden tests, in which variants are collapsed together, can be more powerful if the associated variants have similar effect sizes[3]. To further improve power, we incorporated data on family history of breast cancer (Methods)[4]. Association tests were conducted for all genes in which there was at least one carrier of a variant (15,616 genes for PTVs and 18,601 genes for rare missense variants).

**Fig. 1 | Manhattan plot of $z$ scores from the meta-analysis assessing the association between protein-truncating variant carriers within genes and breast cancer risk.** The $x$ axis is the chromosomal position, and the $y$ axis is the meta-analyzed $z$ score from testing $H_0$: $\beta = \ln(\text{OR}) = 0$ in the UK Biobank and BCAC datasets (two-tailed). The blue lines correspond to $z = \pm 3.29$, $P = 0.001$, the red lines correspond to $z = \pm 4.71$, $P = 2.5 \times 10^{-6}$. All labeled genes are those with $P < 0.001$. All $P$ values are unadjusted for multiple testing.

In the PTV meta-analysis, 30 genes were associated at $P < 0.001$ (Supplementary Table 3 and Figs. 1 and 2). Of these, six met exome-wide significance ($P < 2.5 \times 10^{-6}$), of which five are known breast cancer risk genes—*ATM*, *BRCA1*, *BRCA2*, *CHEK2* and *PALB2*. Associations were also identified for PTVs in *MAP3K1* ($P = 1.2 \times 10^{-9}$). Associations at $P < 1 \times 10^{-4}$ were also identified for PTVs in *LZTR1*, ATR interacting protein (*ATRIP*) and the known risk gene *BARD1*. Of the other previously identified breast cancer susceptibility genes, associations with $P < 0.01$ were observed for *CDH1* and *RAD51D* (Supplementary Table 4). Associations significant at $P < 0.01$ were not observed for other known susceptibility genes, but PTV frequencies were very low and the confidence limits include the previous odds ratio (OR) estimates[1,5].

There was no evidence for an excess of associations significant at $P < 0.001$ after allowing for the six exome-wide significant genes (Fig. 2). However, 28 of the 30 associations at $P < 0.001$ correspond to an increased risk, compared with ~15 that would be expected by chance. This imbalance suggests some of the other associations may be genuine.

We performed additional analyses by age and (within the BCAC dataset) the following disease subtypes: estrogen receptor (ER)$^+$ or ER$^-$, progesterone receptor (PR)$^+$ or PR$^-$ and triple-negative disease. When restricting the age of cases to <50 years, 40 genes were associated (all with increased risk) at $P < 0.001$, suggesting an enrichment of associations in this age group (Supplementary Table 5), *MGAT5* met exome-wide significance, in addition to *BRCA2*, *BRCA1*, *CHEK2*, *PALB2*, *ATM* and *MAP3K1*. In the subtype-specific analyses (Supplementary Table 6), the expected associations for known genes were observed, importantly, the higher OR for ER$^-$ and triple-negative disease for *BRCA1* and higher OR for ER$^+$ disease for *CHEK2*, but no other genes were associated with subtype-specific disease at exome-wide significance.

For the rare missense variant meta-analysis, 28 genes had a $P < 0.001$, 18 of which corresponded to an increased risk of breast cancer (Supplementary Table 7 and Extended Data Figs. 1 and 2) compared to 14 expected by chance. Only *CHEK2* met exome-wide significance ($P = 7.0 \times 10^{-19}$). Associations with $P < 1 \times 10^{-4}$ were also observed for rare missense variants in *SAMHD1*, *HCN2*, *CLIC6* and *ACTL8*.

We next considered missense variants predicted deleterious combined with PTVs. We defined deleterious missense variants using Combined Annotation Dependent Depletion (CADD score > 20) (ref. 6) and Helix (Helix score > 0.5) (ref. 7). When using CADD, 33 genes had a $P < 0.001$, 22 of which corresponded to an increased risk of breast cancer (Supplementary Table 8 and Extended Data Figs. 3 and 4). Six genes met exome-wide significance, including the following known five risk genes: *CHEK2* ($P = 2.8 \times 10^{-66}$), *BRCA2* ($P = 7.2 \times 10^{-44}$), *PALB2* ($P = 4.5 \times 10^{-26}$), *ATM* ($P = 3.3 \times 10^{-21}$) and *BRCA1* ($P = 1.6 \times 10^{-17}$), together with *CDKN2A* ($P = 8.3 \times 10^{-7}$). Associations with $P < 1 \times 10^{-4}$ were also observed for *SAMHD1*, *MRPL27*, *EXOC4* and *PPP1R3B*. When instead defining deleterious rare missense variants using Helix and combining with PTVs, 29 genes had a $P < 0.001$, 25 of which corresponded to an increased risk of breast cancer (Supplementary Table 9). Only the known five genes met exome-wide significance. Associations with $P < 1 \times 10^{-4}$ were also observed for *LZTR1*, *MAP3K1*, *DCLK1*, *MDM4*, *STX3* and *ATRIP*.

Notably, of the genes with $P < 1 \times 10^{-4}$, *MAP3K1*, *LZTR1*, *ATRIP*, *CDKN2A* and *SAMHD1* have prior evidence of being tumor suppressor genes (TSGs). *MAP3K1* is a stress-induced serine/threonine kinase that activates the extracellular signal-regulated kinase (ERK) and Jun N-terminal kinase (JNK) pathways by phosphorylation of MAP2K1 and MAP2K4 (refs. 8,9). Inactivating variants in *MAP3K1* are one of the commonest somatic driver events in breast tumors[10,11]. *MAP3K1* is also a well-established breast cancer GWAS locus[12]; at least three independent signals have been identified mapping to regulatory regions with *MAP3K1* expression as the likely target[13,14]. To evaluate whether the *MAP3K1* PTV association we observed was driven by the GWAS associations, or vice-versa, we fitted logistic regression models to UKB data in which the PTV burden variable and the lead GWAS SNPs (SNP$_1$: rs62355902, SNP$_2$: rs984113 and SNP$_3$: rs112497245) were considered jointly (Supplementary Table 10). In the model with all variables, the OR associated with carrying a PTV (OR = 4.95 (2.27, 10.82)) was similar to the unadjusted OR. Similarly, the ORs for each of the SNPs were similar to the ORs without adjustment for PTVs. This suggests that the PTV burden and GWAS associations are independent and reflect the distinct effects of inactivating coding alterations and regulatory variants.

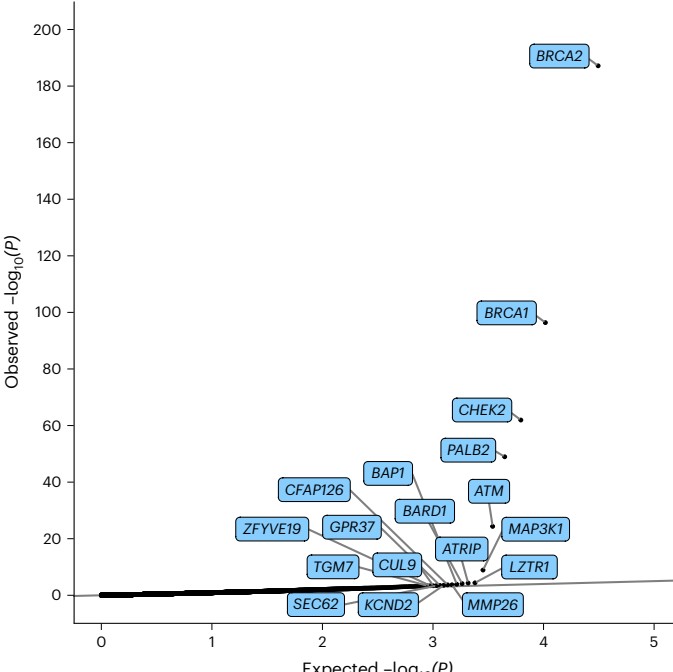

**Fig. 2 | Quantile–quantile plot of P values from the meta-analysis assessing the association between protein-truncating variant carriers and breast cancer risk.** P values are from the meta-analyzed z score from testing $H_0$: $\beta = \ln(\text{OR}) = 0$ in the UKB and BCAC datasets (two-tailed). The x axis is the expected $\log_{10} P$ values from the null hypothesis, the y axis is the observed $\log_{10} P$ values. All highlighted genes have $P < 0.0005$ and are associated with an increased risk of breast cancer. All P values are unadjusted for multiple testing.

*ATRIP* codes for a DNA damage response protein that forms a complex with ATR. ATR–ATRIP is involved in the process that activates checkpoint signaling when single-stranded DNA is detected following the processing of DNA double-stranded breaks or stalled replication forks[15,16]. *LZTR1* codes for a protein found in the Golgi apparatus[17]. Germline mutations in *LZTR1* have been associated with schwannomatosis, a rare tumor predisposition syndrome[18,19]. *CDKN2A* also codes for tumor suppressor proteins, including p16(INK4A) and p14(ARF) (ref. 20). *CDKN2A* is a known melanoma[21] and pancreatic cancer susceptibility gene and is an important TSG altered in a wide variety of tumors, including breast cancer[22]. There have been some previous suggestions that deleterious germline *CDKN2A* is associated with breast cancer risk[23,24]. *SAMHD1* promotes the degradation of nascent DNA at stalled replication forks, limiting the release of single-stranded DNA[25]. *SAMHD1* also encodes dNTPase that protects cells from viral infections[26] and is frequently mutated in multiple tumor types, including breast cancer. Furthermore, damaging germline variants in *SAMHD1* have recently been associated with delayed age at natural menopause and increased all-cause cancer risk[27]. *MDM4* encodes a p53 repressor overexpressed in a variety of tumors[28] and is also a GWAS locus for triple-negative breast cancer[13,29].

Pathology information was available for cases in the BCAC dataset. We tabulated pathology characteristics for carriers of variants in genes with $P < 1 \times 10^{-4}$ in the meta-analysis of PTVs or the meta-analysis of predicted deleterious (CADD) rare missense variants combined with PTVs (Supplementary Tables 11 and 12). These data suggest a slightly higher proportion of mixed lobular and ductal tumors for *LZTR1* PTV carriers and *MRPL27* deleterious rare missense variant or PTV carriers. There was a slightly higher proportion diagnosed >50 years for *ATRIP* PTV carriers and a higher proportion of HER2+ tumors for *EXOC4* deleterious rare missense

variant or PTV carriers. However, the number of carriers is small in each case.

We performed Gene Set Enrichment Analysis (GSEA) based on the PTV associations for pathways in the Kyoto Encyclopedia of Genes and Genomes (KEGG), BioCarta and Reactome. We did this for all genes including and excluding *BRCA1*, *BRCA2*, *ATM*, *CHEK2* and *PALB2*. When including the five genes, 28 pathways had a false discovery rate ($q$) < 0.05 (Supplementary Table 13). Of these, all but one (Reactome peptide hormone biosynthesis) include *BRCA1* or *BRCA2*. The top pathway was Reactome DNA double-strand break repair. After excluding the five genes (Supplementary Table 14), the strongest enrichment was for the BioCarta NFKB and CD40 pathways (which contain *MAP3K1*), Reactome DNA double-strand break response and Reactome hormone peptide biosynthesis (all $q$ < 0.10).

To evaluate the overall contribution of PTVs to the FRR, we fitted models to the effect size using an empirical Bayes approach. We used whole-genome sequencing data in UKB to estimate the missing contribution due to large rearrangements. Under the assumption of an exponentially distributed effect size, the estimated proportion of risk genes ($a$) was 0.0047 with a median OR of 1.38. Under this model, an estimated 10.61% of the FRR would be explained by all PTVs, of which 9.64% would be due to the five genes *BRCA1*, *BRCA2*, *ATM*, *CHEK2* and *PALB2* and 0.97% due to all other genes combined with *MAP3K1* contributing 0.14% (Supplementary Table 15). Only the six genes reaching exome-wide significance for PTVs had a posterior probability of association >0.90. We repeated these analyses using the subsets of genes including breast cancer driver genes and target genes of GWAS signals identified in ref. 13, the list of cancer predisposition genes identified in ref. 30, Catalogue Of Somatic Mutations In Cancer (COSMIC) TSGs[31] and the top pathways identified by GSEA (Supplementary Table 16). The largest contributions to the FRR, after excluding the five known genes, were for the BioCarta CD40 pathway (0.657%, $n = 16$, $a = 0.628$) and COSMIC TSGs (0.639%, $n = 320$, $a = 0.196$). Thus, these results suggest that the majority of the remaining risk genes are TSGs.

These results demonstrate that large exome sequencing studies, combined with efficient burden analyses, can identify additional breast cancer susceptibility genes. The excess of positive associations at $P < 0.001$ indicates that further genes should be identifiable through large datasets—the heritability analyses suggest the number of associated genes might be of the order of 90, with the majority of these being TSGs. Although the estimated risks associated with the new genes, in particular *MAP3K1* PTVs, would be large enough to be of clinical relevance, the effect sizes might be over-estimated due to the 'winner's curse'[32]. Thus, further replication in larger datasets will also be necessary to provide more precise estimates for variants in the new genes, to define the set of variants in these genes associated with breast cancer, the clinic-pathological characteristics of tumors in variant carriers and the combined effects of pathogenic variants and other risk factors. The heritability analyses suggest that most of the contribution of PTVs is mediated through the five genes *BRCA1*, *BRCA2*, *ATM*, *CHEK2* and *PALB2*, commonly tested for in clinical cancer genetics[33]. These analyses assume dominant inheritance and recessive genes may also contribute to the familial risk, while subsets of missense variants may also make important contributions (exemplified by *CDKN2A* and *SAMHD1*). However, these results suggest that the majority of the 'missing' heritability is likely to be found in the noncoding genome.

## Online content

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

**Naomi Wilcox**[1], **Martine Dumont**[2], **Anna González-Neira**[3], **Sara Carvalho**[1], **Charles Joly Beauparlant**[2], **Marco Crotti**[1], **Craig Luccarini**[4], **Penny Soucy**[2], **Stéphane Dubois**[2], **Rocio Nuñez-Torres**[3], **Guillermo Pita**[3], **Eugene J. Gardner** [5], **Joe Dennis**[1], **M. Rosario Alonso**[3], **Nuria Álvarez**[3], **Caroline Baynes**[4], **Annie Claude Collin-Deschesnes**[2], **Sylvie Desjardins**[2], **Heiko Becher** [6], **Sabine Behrens** [7], **Manjeet K. Bolla**[1], **Jose E. Castelao**[8], **Jenny Chang-Claude**[7,9], **Sten Cornelissen**[10], **Thilo Dörk** [11], **Christoph Engel**[12,13], **Manuela Gago-Dominguez**[14], **Pascal Guénel** [15], **Andreas Hadjisavvas**[16],

Eric Hahnen[17,18], Mikael Hartman[19,20,21], Belén Herráez[3], SGBCC Investigators*, Audrey Jung[7], Renske Keeman[10], Marion Kiechle[22], Jingmei Li [23] ✉, Maria A. Loizidou[16], Michael Lush [1], Kyriaki Michailidou [1,24], Mihalis I. Panayiotidis[16], Xueling Sim [19], Soo Hwang Teo [25,26], Jonathan P. Tyrer [4], Lizet E. van der Kolk[27], Cecilia Wahlström[28], Qin Wang [1], John R. B. Perry [5,29], Javier Benitez[30,31], Marjanka K. Schmidt [10,32], Rita K. Schmutzler[17,18,33], Paul D. P. Pharoah [1,4], Arnaud Droit[2,34], Alison M. Dunning [4], Anders Kvist [28], Peter Devilee [35,36], Douglas F. Easton [1,4,45] ✉ & Jacques Simard [2,45]

[1]Centre for Cancer Genetic Epidemiology, Department of Public Health and Primary Care, University of Cambridge, Cambridge, UK. [2]Genomics Center, Centre Hospitalier Universitaire de Québec—Université Laval Research Center, Québec City, Quebec, Canada. [3]Human Genotyping Unit-CeGen, Human Cancer Genetics Programme, Spanish National Cancer Research Centre (CNIO), Madrid, Spain. [4]Centre for Cancer Genetic Epidemiology, Department of Oncology, University of Cambridge, Cambridge, UK. [5]MRC Epidemiology Unit, Wellcome-MRC Institute of Metabolic Science, University of Cambridge, Cambridge, UK. [6]Institute of Medical Biometry and Epidemiology, University Medical Center Hamburg-Eppendorf, Hamburg, Germany. [7]Division of Cancer Epidemiology, German Cancer Research Center (DKFZ), Heidelberg, Germany. [8]Oncology and Genetics Unit, Instituto de Investigación Sanitaria Galicia Sur (IISGS), Xerencia de Xestion Integrada de Vigo-SERGAS, Vigo, Spain. [9]Cancer Epidemiology Group, University Cancer Center Hamburg (UCCH), University Medical Center Hamburg-Eppendorf, Hamburg, Germany. [10]Division of Molecular Pathology, The Netherlands Cancer Institute, Amsterdam, the Netherlands. [11]Gynaecology Research Unit, Hannover Medical School, Hannover, Germany. [12]Institute for Medical Informatics, Statistics and Epidemiology, University of Leipzig, Leipzig, Germany. [13]LIFE—Leipzig Research Centre for Civilization Diseases, University of Leipzig, Leipzig, Germany. [14]Cancer Genetics and Epidemiology Group, Instituto de Investigación Sanitaria de Santiago de Compostela (IDIS) Foundation, Complejo Hospitalario Universitario de Santiago, SERGAS, Santiago de Compostela, Spain. [15]Team 'Exposome and Heredity,' CESP, Gustave Roussy, INSERM, University Paris-Saclay, UVSQ, Villejuif, France. [16]Department of Cancer Genetics, Therapeutics and Ultrastructural Pathology, The Cyprus Institute of Neurology & Genetics, Nicosia, Cyprus. [17]Center for Familial Breast and Ovarian Cancer, Faculty of Medicine and University Hospital Cologne, University of Cologne, Cologne, Germany. [18]Center for Integrated Oncology (CIO), Faculty of Medicine and University Hospital Cologne, University of Cologne, Cologne, Germany. [19]Saw Swee Hock School of Public Health, National University of Singapore and National University Health System, Singapore City, Singapore. [20]Department of Surgery, National University Health System, Singapore City, Singapore. [21]Department of Pathology, Yong Loo Lin School of Medicine, National University of Singapore, Singapore City, Singapore. [22]Division of Gynaecology and Obstetrics, Klinikum rechts der Isar der Technischen Universität München, Munich, Germany. [23]Genome Institute of Singapore, Agency for Science, Technology and Research, Singapore City, Singapore. [24]Biostatistics Unit, The Cyprus Institute of Neurology & Genetics, Nicosia, Cyprus. [25]Breast Cancer Research Programme, Cancer Research Malaysia, Subang Jaya, Malaysia. [26]Department of Surgery, Faculty of Medicine, University of Malaya, UM Cancer Research Institute, Kuala Lumpur, Malaysia. [27]Family Cancer Clinic, The Netherlands Cancer Institute—Antoni van Leeuwenhoek hospital, Amsterdam, the Netherlands. [28]Division of Oncology, Department of Clinical Sciences Lund, Lund University, Lund, Sweden. [29]Metabolic Research Laboratory, Wellcome-MRC Institute of Metabolic Science, University of Cambridge, Cambridge, UK. [30]Human Genetics Group, Spanish National Cancer Research Centre (CNIO), Madrid, Spain. [31]Centre for Biomedical Network Research on Rare Diseases (CIBERER), Instituto de Salud Carlos III, Madrid, Spain. [32]Division of Psychosocial Research and Epidemiology, The Netherlands Cancer Institute—Antoni van Leeuwenhoek hospital, Amsterdam, the Netherlands. [33]Center for Molecular Medicine Cologne (CMMC), Faculty of Medicine and University Hospital Cologne, University of Cologne, Cologne, Germany. [34]Département de Médecine Moléculaire, Faculté de Médecine, Centre Hospitalier Universitaire de Québec Research Center, Laval University, Québec City, Quebec, Canada. [35]Department of Pathology, Leiden University Medical Center, Leiden, the Netherlands. [36]Department of Human Genetics, Leiden University Medical Center, Leiden, the Netherlands. [45]These authors jointly supervised this work: Douglas F. Easton, Jacques Simard. *A list of authors and their affiliations appears at the end of the paper. ✉e-mail: lijm1@gis.a-star.edu.sg; dfe20@medschl.cam.ac.uk

## SGBCC Investigators

Benita Kiat-Tee Tan[37,38,39], Veronique Kiak Mien Tan[37,38], Su-Ming Tan[40], Geok Hoon Lim[41], Ern Yu Tan[42,43,44], Peh Joo Ho[23] & Alexis Jiaying Khng[23]

[37]Department of Breast Surgery, Singapore General Hospital, Singapore City, Singapore. [38]Division of Surgical Oncology, National Cancer Centre, Singapore City, Singapore. [39]Department of General Surgery, Sengkang General Hospital, Singapore City, Singapore. [40]Division of Breast Surgery, Department of General Surgery, Changi General Hospital, Singapore City, Singapore. [41]KK Breast Department, KK Women's and Children's Hospital, Singapore City, Singapore. [42]Department of General Surgery, Tan Tock Seng Hospital, Singapore City, Singapore. [43]Lee Kong Chian School of Medicine, Nanyang Technological University, Singapore City, Singapore. [44]Institute of Molecular and Cell Biology, Singapore City, Singapore.

## Methods

### UKB

The UKB is a population-based prospective cohort study of more than 500,000 subjects. More detailed information on the UKB is given elsewhere[34,35]. The study received ethics approval from the North West Multi-center Research Ethics Committee. All participants signed written informed consent before participating. WES data for 450,000 subjects were released in October 2021 and accessed via the UKB DNA Nexus platform[36]. QC metrics were applied to Variant Call Format (VCF) files as discussed in ref. 37, including genotype level filters for depth and genotype quality.

Samples with disagreement between genetically determined and self-reported sex, sex aneuploidy or excess relatives in the dataset were excluded. Excess relatives were identified by considering pairs of individuals with kinship >0.17. If one individual in a pair was a case and one was a control then the case was preferentially selected; otherwise, one individual was selected at random. Genetic ancestry was estimated using genetic principal components and the Gilbert–Johnson–Keerthi distance algorithm[38]. If genetic principal components were not available, self-reported ancestry was used. Samples of ancestry other than European were excluded. The final dataset for analysis included 419,307 samples with 227,393 females. Cases were defined by having invasive breast cancer (International Classification of Diseases (ICD)-10 code C50) or carcinoma in situ (D05), as determined by linkage to the National Cancer Registration and Analysis Service (NCRAS), or self-reported breast cancer. Both prevalent and incident cases were included. Only breast cancers that were an individual's first or second diagnosed cancer were included as cases. By this definition, 17,958 female and 94 male cases were included.

For structural variants, we accessed the structural variant population VCF files for the initial release of UKB whole-genome sequencing via the DNA Nexus platform. These deletions, duplications and insertions were called using GraphTyper (2.7.1) (refs. 39,40). We identified any structural variant that passed the GraphTyper QC filters and overlapped an exon of the MANE transcripts of each gene. The samples were filtered using the above exclusions leaving 64,264 samples (4,847 female breast cancer cases and 59,417 female controls). The frequency of structural variants was then calculated for each gene and used to adjust the PTV frequency (Supplementary Methods).

### The BCAC datasets

The BRIDGES and PERSPECTIVE samples were from studies in the BCAC (BRIDGES: eight studies, PERSPECTIVE: three studies; Supplementary Table 1). All studies were approved by ethical review boards (Supplementary Table 17). All subjects provided written informed consent. Most samples were previously included in a targeted panel sequencing project[1]. Phenotype data were based on the BCAC database v14. Samples were oversampled for early-onset (age of diagnosis below 50 years) or family history of breast cancer. Cases were preferentially selected to have information on tumor pathology. Samples with previously identified pathogenic mutations in *BRCA1*, *BRCA2* or *PALB2* (348 cases, 176 controls) were not included.

For BRIDGES, library preparation was conducted in the three laboratories using the Nextera DNA Exome kit (Illumina) for tagmentation, barcoding and amplification steps. Subsequently, 500 ng of DNA per sample was pooled in 12-plex and concentrated using a vacuum system. Afterward, hybridization capture reagents for DNA libraries were used for overnight hybridization with the xGen Exome Research panel (Integrated DNA Technologies), capture and amplification. Barcoded pooled libraries of 96 samples were sequenced on each lane of a NovaSeq 6000 S4 flowcell (Illumina) using NovaSeq XP 4-Lane Kit (2 × 100 bp).

For PERSPECTIVE, library preparation was conducted using Agilent SureSelect Human all exon V7 (48.2 Mb). Barcoded libraries of 88 samples were sequenced on a NovaSeq 6000 S4 flowcell (Illumina) using NovaSeq XP 4-Lane Kit (2 × 100 bp).

The same pipeline for variant calling was applied to both the BRIDGES and PERSPECTIVE data and followed the Genome Analysis Toolkit (GATK) best practices. Briefly, raw sequence data (FASTQ format) were preprocessed to produce BAM files. This involved alignment to the reference genome (hg38) using BWA (v0.7.17) and the sorting and indexing of the reads using samtools (v1.10). Identification and removal of duplicate read pairs from the same DNA fragments were performed using Picard's MarkDuplicates (v2.1.1). Base recalibration included the generation of a base quality score recalibration table with the GATK BaseRecalibrator software (v4.1.4.1), later applied to the read bases to adjust their quality scores and increase the accuracy of the variant calling algorithms with the GATK BQSR (v4.1.4.1). An intermediate and informal QC was performed for a sanity check, including coverage and alignment mapping metrics using samtools flagstat software (v1.10) and Picard (v2.22.2). Variants were then called using GATK HaplotypeCaller (v4.1.4.1). The GATK GenotypeGVCFs (v4.1.4.1) tool was used for the joint genotyping step on each genomic database. The variants with excess heterozygosity were filtered out using GATK VariantFiltration (v4.1.4.1) and GATK MakeSitesOnlyVcf (v4.1.4.1). The GATK VariantRecalibrator (v4.1.4.1) software was used to produce tranches files on SNPs and indels separately. Finally, the tranches files were used to apply the recalibration using the GATK ApplyVQSR (v4.1.4.1). Further details are provided in Supplementary Methods.

For the final dataset, similar QC filtering as for the UKB was applied, using VCFtools (v0.1.15), BCFtools (v1.9), Picard (v2.22.2) and PLINK (v1.90b). At the genotype level, SNPs were excluded with sequencing depth <7 or heterozygous allele balance <0.15 or >0.85. Indels were excluded with sequencing depth <10 or allele balance <0.2 or >0.8. On males' X chromosomes, depth filters were reduced to 5 for SNPs and 7 for indels. Samples with missing calls for >15% and variants with missing calls for >15% of samples were excluded. Variants with Hardy–Weinberg equilibrium $P$ value $10^{-15}$ were also removed. We also excluded samples where the genotypes were inconsistent with previous array genotyping or targeted sequencing data[1,2].

The BRIDGES study sequenced 6,912 samples, of which 3,461 cases and 3,200 controls remained in the final dataset after QC. The PERSPECTIVE study sequenced 10,523 samples, of which 4,777 cases and 5,210 remained in the final dataset.

### Data preparation

For both the UKB and BCAC datasets, Ensembl Variant Effect Predictor (VEP) v101.0 was used to annotate variants[41]. Annotations included the 1000 genomes phase 3 allele frequency, sequence ontology variant consequences and exon/intron number. For each gene, the MANE Select[42] transcript was used if it was available for that gene, or the RefSeq Select transcript[43]. Annotation files were used to identify PTVs and rare (allele frequency <0.001 in both the 1000 genomes dataset and the current dataset) missense variants. PTVs in the last exon of each gene and the last 50 bp of the penultimate exon were excluded as these are generally predicted to escape nonsense-mediated mRNA decay. VEP was also used to annotate missense variants by CADD score (v1.6) (ref. 6). Here CADD ≥ 20 was used to define variants predicted to be deleterious. We also defined deleterious missense variants using Helix scores (v4.4.1) (ref. 7).

### Burden test analysis

Association analyses were carried out for each gene separately for PTVs, rare missense variants and predicted deleterious rare missense variants (defined by CADD score ≥ 20 or Helix score > 0.5) and PTVs combined. The main association analyses were burden tests in which genotypes were collapsed to a 0/1 variable based on whether samples carried a variant of the given class. That is, $G_i = 1$ if $\sum_{j=1}^{p} g_{ij} > 0$ and 0 if $\sum_{j=1}^{p} g_{ij} = 0$, where $g_{ij} = 0, 1, 2$ is the number of minor alleles observed for sample $i$ at variant $j$, and $p$ is the number of variants in the gene (thus, heterozygous and homozygous carriers were combined). All $P$ values were two-sided.

We used logistic regression analysis to test for an association between carriers of variants within a gene and breast cancer status. We incorporated family history as a surrogate for disease status, similar to the method presented in ref. 44. This markedly improves power because susceptibility variants will also be associated with family history; in particular, it allows information on males in the cohort with a family history of female breast cancer to be used. To do this, we treated genotype (0/1) as the dependent variable and family history weighted disease status as the covariate; the latter is defined as $d + 1/2 f$, where $d = 0, 1$ was the disease status of the genotyped individual and $f = 0$ or 1 according to whether the individual reported a positive first-degree family history. The rationale for this weighting is that, for small effect sizes, the log-OR associated with a positive first-degree relative is approximately ½ that associated with the disease. (The approach of using family history as a surrogate was suggested in ref. 44. This method differs in that family history is included with a weight of ½ rather than 1, or using a 2-degree freedom test.) For the BCAC dataset, we adjusted for country and library preparation method (BRIDGES versus PERSPECTIVE). Ancestry was not adjusted for as within each country ancestry was constant. For the UKB dataset, we adjusted for the first ten principal components and sex. For genes on chromosome X, only females were used in the analysis. When looking at case–control analyses for subtypes of the disease, for example ER status, in the BCAC dataset logistic regression was also used.

*NUDT11* was excluded because missing genotypes (which were treated as noncarriers) led to spurious associations, although the variants passed QC filtering. *AFF1* was also removed as the PTV frequency was high in PERSPECTIVE but rare in BRIDGES and UKB. This was likely due to a single PTV artifact within the PERSPECTIVE dataset.

To combine the results from the BCAC and UKB datasets in a meta-analysis, the association tests for each gene were converted to $z$ scores. The combined $z$ score was defined as $z_M = \frac{\sum_j w_j z_j}{\sqrt{w_j^2}}$. Here $z_M$ is the combined $z$ score, $z_j$ is the $z$ score for study $j$ and $w_j$ is the weight associated with study $j$.

A standard meta-analysis would define the weights $w_j$ using inverse variance or effective sample sizes. However, the effect sizes from the BCAC and UKB may not be comparable, because the BCAC studies oversampled for family history and early age at onset, which may have increased the estimated effect. Therefore, we defined weights by using the associations in the known risk gene *CHEK2* as a standard—we rationalized that the *CHEK2* PTVs provided the best standard as the association is well-established and the OR is highly reproducible[1,5,45]. Moreover, the OR (~2 to 3) was representative of the size of effects we hoped to detect for other genes. Thus, we defined $(w_1, w_2) = \left(\frac{z_1}{z_2}, 1\right)$, where $\frac{z_1}{z_2}$ is the ratio of $z$ scores for *CHEK2* for the BCAC dataset and UKB. The approach is equivalent to a meta-analysis of risk per unit dose in studies with different levels of exposure or dose (with dose here being the log-OR for *CHEK2*)[46,47]. As a sensitivity analysis, we also derived weightings based on a combined analysis of the five known genes *ATM, BRCA1, BRCA2, CHEK2* and *PALB2*. This gives slightly more weight to UKB (BCAC: UKB 0.307 versus 0.473) but does not change the genes reaching exome-wide significance (the ten most significant genes for PTVs were identical; Supplementary Table 18). The same weights were applied for the meta-analysis of the other variant categories. The $z_M$ scores were plotted in Manhattan plots, and associated $P$ values were plotted in quantile–quantile plots. For PTVs, the lambda statistic for inflation in the test statistics (based on the median chi-squared statistic) was 0.766 for UKB, 0.688 for BCAC and 0.725 for the meta-analysis, indicating that the tests were somewhat conservative on average.

We compared this approach to the method outlined in ref. 48. This method is similar to a random-effect meta-analysis but assumes no heterogeneity under the null hypothesis. When heterogeneity is present, this method can achieve greater power than traditional random-effect methods that do not normally achieve greater power than fixed-effect methods. We tested this method for genes with $P < 1 \times 10^{-4}$ from the PTV meta-analysis as described above. $P$ values using this method were only smaller for the genes *BRCA1, BRCA2* and *PALB2*. Furthermore, tau, the estimated amount of total heterogeneity, was estimated to be 0 for all genes apart from *BRCA1* and *BRCA2*, suggesting that for most genes this method is not an improvement to the method above using *CHEK2* PTV $z$ scores as weights in a fixed-effect approach (Supplementary Table 19).

To investigate the joint effect of PTVs in *MAP3K1* and common susceptibility variants in the region identified through GWAS, we accessed imputed genotype data from UKB for the lead SNPs as identified through previous fine-mapping analyses[13,14]. We fitted logistic regression models including covariates for PTVs and the lead SNPs and compared the fit of the model and effect sizes, with the model in which the PTVs or the lead SNPs were excluded.

Data on clinicopathological characteristics of cases in the BCAC dataset were also accessed, and the proportion of individuals with specific pathologic features, for example stage and grade, were compared between carriers of variants in a specific gene, for example, *MAP3K1* PTV carriers, and the overall dataset.

## Pathway analysis
We performed GSEA[49] to evaluate the enrichment of genes in KEGG[50], Reactome[51] and BioCarta[52] pathways in breast cancer risk using the R package clusterProfiler[53]. Pathway lists were accessed using MSigDB[54,55]. We used $z$ scores from the PTV meta-analysis to create an ordered gene list. We did this using all genes in each pathway and excluding *BRCA1, BRCA2, CHEK2, PALB2* and *ATM*. Results were presented in terms of false discovery rates ($q$).

## Contribution of PTVs to the FRR
We estimated the overall contribution of PTVs to the FRR of breast cancer using an empirical Bayesian approach. Given the aggregate frequency $p_j$ of PTVs in a gene is rare, and all PTVs confer the same relative risk $e^{\beta_j}$, the FRR due to one gene, given $p_j$ and $e^{\beta_j}$, is[1]

$$\lambda_j = 1 + \frac{p_j(e^{\beta_j} - 1)^2}{\left(2p_j\left(e^{\beta_j} - 1\right) + 1\right)^2}$$

Under the additional assumption that the risks conferred by variants in different genes are additive, the total contribution over $J$ genes is given by

$$\lambda = 1 + \sum_{j=1}^{J} (\lambda_j - 1)$$

We ignored recessive effects in this analysis—because PTV homozygotes are extremely rare for most genes their effect is difficult to estimate. These results can therefore be interpreted as the contribution to the FRR to the offspring of affected individuals. However, there is limited evidence of a higher familial risk of breast cancer to siblings that would indicate an important rare recessive component. We assumed a prior distribution for effect sizes (log-OR) in which a proportion $\alpha$ of genes are risk associated and the estimated log-OR, $\beta$, for associated genes have an exponential distribution with parameter $\eta$; this distribution was chosen because the distribution of effect sizes is likely to be skewed, with only a small number of genes have a large effect size and most undiscovered genes having smaller effect sizes. An approximate likelihood of the observed carrier count data, by gene, was derived, summed over all genes and maximized numerically to estimate $\alpha$ and $\eta$, and hence posterior effect size distributions given the data. We estimated $p_j$ for each gene using the PTV carrier counts and then updated this to additionally account for the structural variant frequency in the

gene. The total contribution to the FRR was estimated by integrating the FRR estimates given $\beta_j$ over the posterior distribution. Further details for the methods are given in Supplementary Methods.

We calculated the contribution of PTVs to the FRR for all genes in the dataset and also for subsets of genes including breast cancer driver genes and target genes of GWAS signals identified in ref. 13, the list of cancer predisposition genes identified in ref. 30, COSMIC TSGs[31] and the top pathways identified by GSEA.

## Statistics and reproducibility
No statistical method was used to predetermine the sample size. The experiments were not randomized, and we did not use blinding. Some samples were excluded for reasons as described in the methods above, for example, for sex discrepancies, excess relatives or discrepancies with previous genotyping. The analyses were conducted as meta-analyses combining the BCAC and UKB datasets.

## Reporting summary
Further information on research design is available in the Nature Portfolio Reporting Summary linked to this article.

## Data availability
Meta-analysis summary statistics are available from the GWAS Catalog (https://www.ebi.ac.uk/gwas/; https://ftp.ebi.ac.uk/pub/databases/gwas/summary_statistics/), accession numbers GCST90267995, GCST90267996, GCST90267997 and GCST90267998. Summary statistics are provided for all ancestries combined as the sample size for non-European ancestry subjects is too small to provide meaningful statistics. Individual level data for the BCAC data are not publicly available due to ethical review board constraints but are available on request through the BCAC Data Access Co-ordinating Committee (BCAC@medschl.cam.ac.uk). Requests for access to UK Biobank data should be made to the UK Biobank Access Management Team (access@ukbiobank.ac.uk).

## Code availability
Quality control filtering of VCF files was performed using VCFtools (v0.1.15), BCFtools (v1.9), Picard (v2.22.2) and PLINK (v1.90b), as outlined in the Methods. Variants were annotated using Ensembl Variant Effect Predictor v101 with assembly GRCh38. The code for each software is available on the website of each package. Data manipulation and analysis were performed using R-4.13 with packages clusterProfiler (4.2.2), data.table (1.14.2), dplyr (1.0.9), gtools (3.9.2.1), HGNChelper (0.8.9), msigdbr (7.5.1), tibble (3.1.7) and tidyr (1.2.0). Plots were created using R-4.13 using additional packages ggplot2 (3.3.6) and ggrepel (0.9.1). The code for each of the R packages can be found in their associated vignettes.

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

## Acknowledgements
Sequencing and analysis for this project were funded by the European Union's Horizon 2020 Research and Innovation Program (BRIDGES: grants 634935 to A.G.-N., A.M.D., A.K., P.D. and D.F.E.), the PERSPECTIVE I&I project (funded by the Government of Canada through Genome Canada and the Canadian Institutes of Health Research, the Ministère de l'Économie et de l'Innovation du Québec through Genome Québec, the Quebec Breast Cancer Foundation, Agilent Technologies Canada and Illumina Canada Ulc to J.S.), the Wellcome Trust (grant v203477/Z/16/Z to S.H.T. and D.F.E.) and the Medical Research Council (unit programs MC_UU_12015/2 and MC_UU_00006/2 to S.H.T.). BCAC is funded by the European Union Horizon 2020 Research and Innovation Program (grants 634935 for BRIDGES and 633784 for B-CAST to M.K.S., P.D.P.P. and D.F.E.), the PERSPECTIVE I&I project and via the Confluence project which is funded with intramural funds from the National Cancer Institute Intramural Research Program, National Institutes of Health (to D.F.E.). The funders had no role in the study design, data collection, data analysis, data interpretation or writing of the report. This research has been conducted using the UKB Resource under application number 28126. BCAC study-specific funding is given in the Supplementary Note. N.W. was supported by the International

Alliance for Cancer Early Detection, an alliance between Cancer Research UK (C14478/A29329), Canary Center at Stanford University, the University of Cambridge, OHSU Knight Cancer Institute, University College London and the University of Manchester.

## Author contributions

A.G.-N., A.M.D., A.K., P.D., D.F.E. and J.S. conceived the study. D.F.E. and J.S. jointly supervised this work. D.F.E. directed the overall analysis. N.W. performed the statistical analysis. M.D., A.G.-N., C.L. and A.K. led the sequencing, with support from P.S., S. Dubois, R.N.-T., M.R.A., N.A., C.B., A.C.C-D., S. Desjardins and C.W. N.W., S.C., C.J.B., M.C., G.P., E.J.G., J.D., M.L., J.P.T., J.D.P. and A.D developed the bioinformatics and computational pipelines. M.K.B., S.B., R.K. and Q.W. led data management within the BCAC. H.B., J.E.C., J.C-C., S.C., T.D., C.E., M.G.-D., P.G., A.H., E.H., M.H., B.H., A.J., M.K., J.L., M.A.L., K.M., M.I.P., X.S., S.H.T., L.E.v.d.K., M.K.S., R.K.S., P.D.P.P. and P.D. contributed to the design and conduct of the contributing BCAC studies. J.C-C., M.K.S. and P.D.P.P. led working groups within the BCAC. N.W. and D.F.E. drafted the manuscript. All authors reviewed and approved the paper.

## Competing interests

E.J.G. and J.R.B.P. hold shares in and are employees of Insmed Inc. All other authors declare no competing interests.

## Additional information

**Extended data** is available for this paper at https://doi.org/10.1038/s41588-023-01466-z.

**Correspondence and requests for materials** should be addressed to Jingmei Li or Douglas F. Easton.

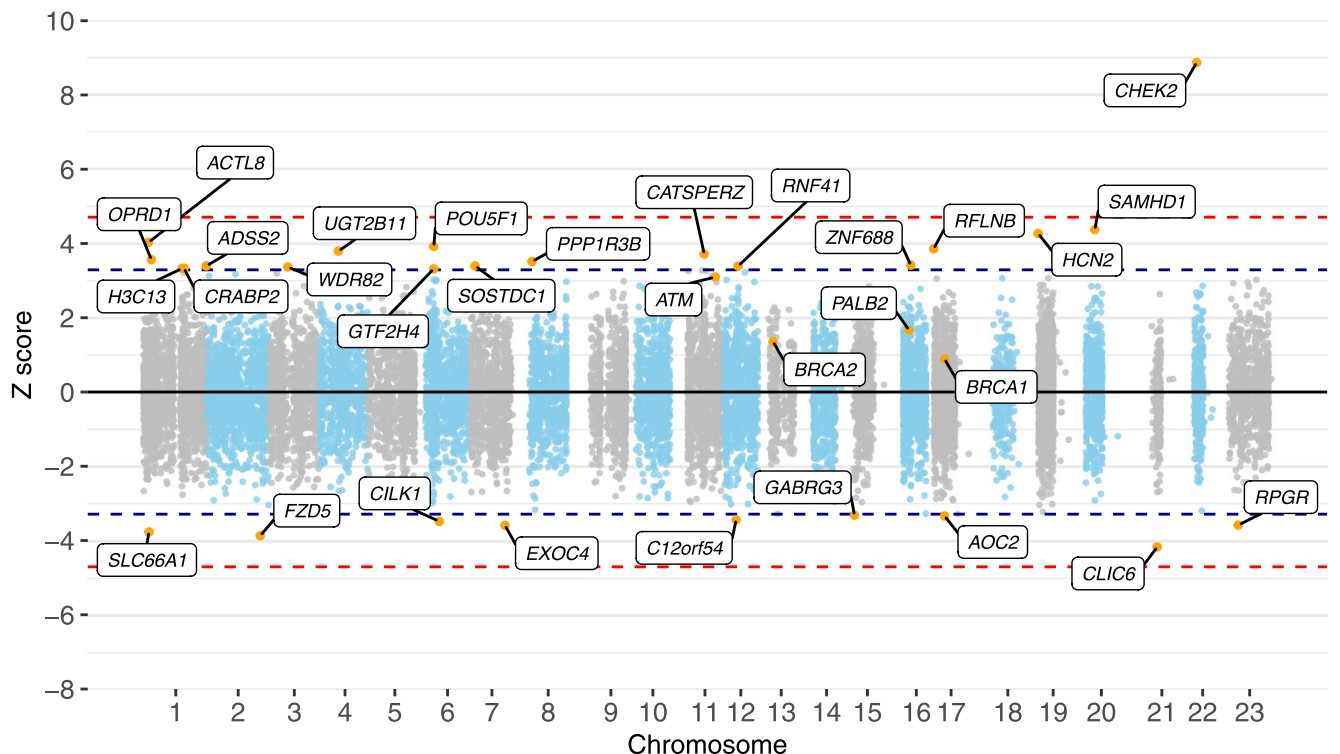

**Extended Data Fig. 1 | Manhattan plot of Z-scores from the meta-analysis assessing the association between rare missense variant carriers by gene and breast cancer risk.** The x-axis gives chromosomal position, and the y values are the meta-analyzed Z-scores from testing $H_0$: $\beta = \ln(OR) = 0$ $\beta = \ln = 0$ in the UKB and BCAC datasets (2-tailed). The blue lines correspond to $Z = \pm 3.29 \pm 3.29$, $P = 0.001$, the red lines correspond to $Z = \pm 4.71 \pm 4.71$, $P = 2.5 \times 10^{-6}$. The labeled genes are those with $P < 0.001$. All $P$-values are unadjusted for multiple testing.

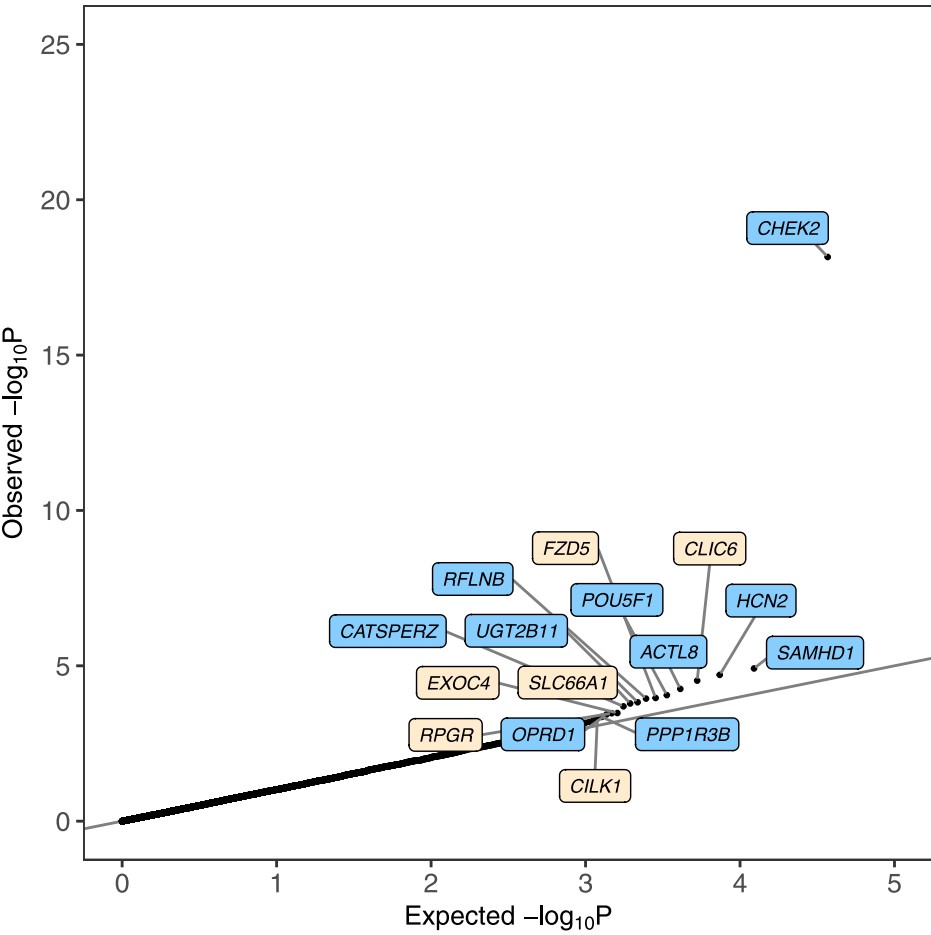

**Extended Data Fig. 2 | Quantile-quantile plot of P-values from the meta-analysis assessing the association between rare missense variant carriers by gene and breast cancer risk.** *P*-values are from the meta-analyzed Z-score from testing H$_0$: $\beta=\ln(OR)=0\beta=\ln=0$ in the UKB and BCAC datasets (2-tailed). The x-axis gives the expected log$_{10}$ P-values under the null hypothesis and the y-axis the observed log$_{10}$ P-values Highlighted genes are those with *P* < 0.0005. Blue corresponds to an increased risk of breast cancer, and cream corresponds to a decreased risk of breast cancer. All *P*-values are unadjusted for multiple testing.

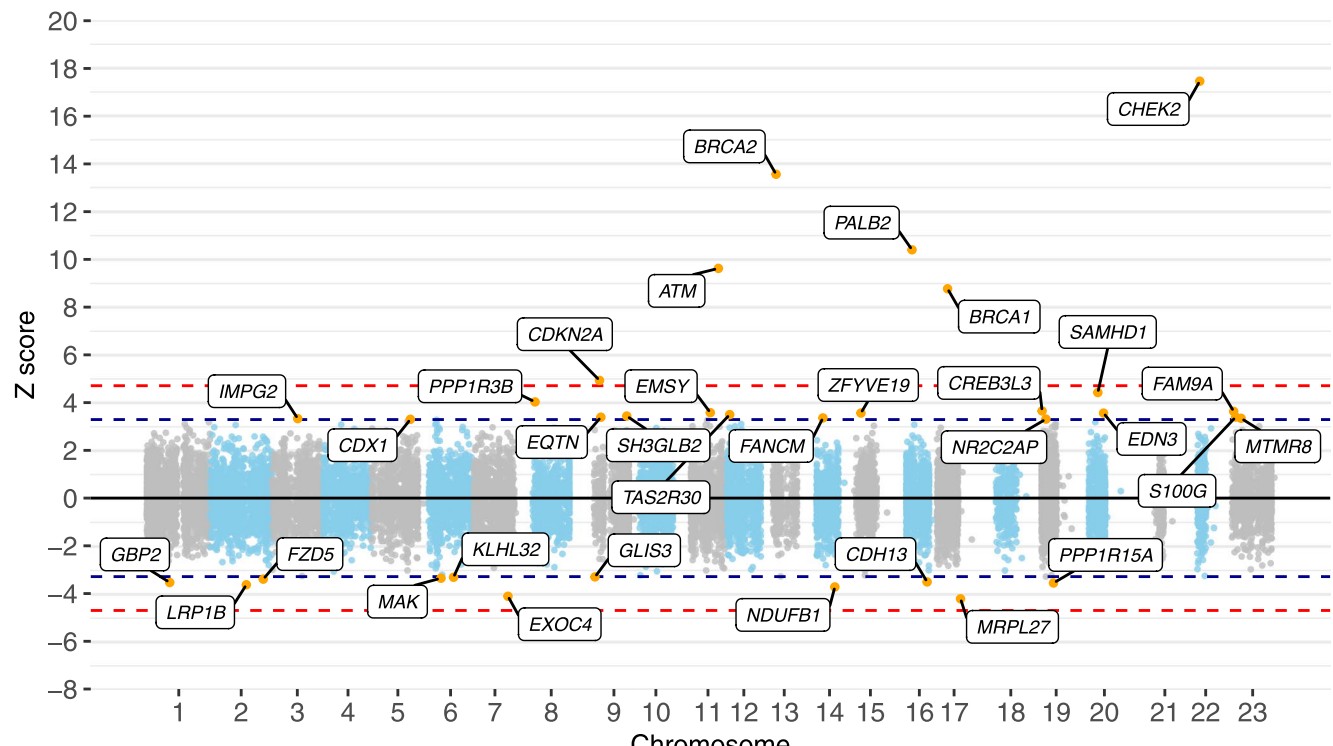

**Extended Data Fig. 3 | Manhattan plot of Z-scores from the meta-analysis assessing the association between PTV or deleterious (CADD > 20) rare missense variant carriers by gene and breast cancer risk.** The x-axis gives the chromosomal position, and the y values are meta-analyzed Z-scores from testing $H_0$: $\beta = \ln(OR) = 0$ $\beta = \ln = 0$ in the UKB and BCAC datasets (2-tailed). The blue lines correspond to Z=±3.29 ± 3.29, $P = 0.001$, the red lines correspond to Z=± 4.71 ± 4.71, $P = 2.5 \times 10^{-6}$. All labeled genes are those with $P < 0.001$. All $P$-values are unadjusted for multiple testing.

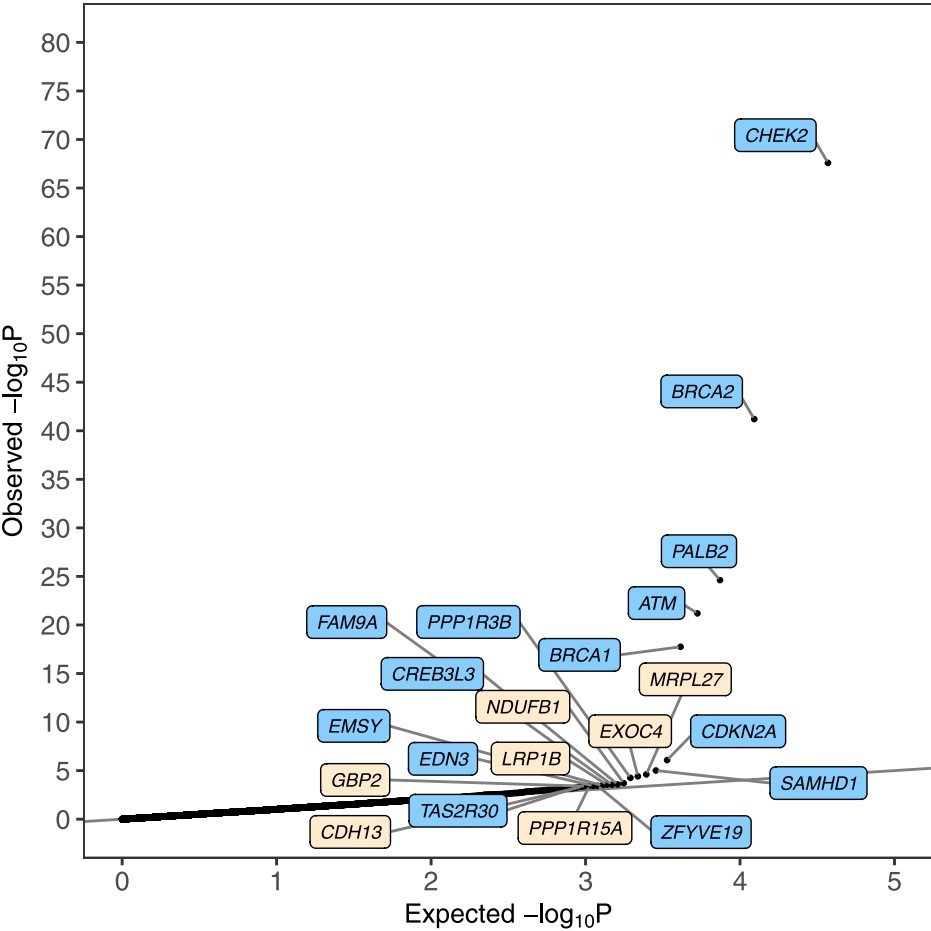

**Extended Data Fig. 4 | Quantile-quantile plot of P-values from the meta-analysis assessing the association between PTV or deleterious (CADD > 20) rare missense variant carriers by gene and breast cancer risk.** *P*-values are from the meta-analyzed Z-score from testing $H_0$: $\beta = \ln(OR)=0$ $\beta = \ln =0$ in the UKB and BCAC datasets (2-tailed). The x-axis gives the expected $\log_{10} P$-values under the null hypothesis and the y-axis the observed $\log_{10} P$-values Highlighted genes are those with $P < 0.0005$. Blue corresponds to an increased risk of breast cancer, and cream corresponds to a decreased risk of breast cancer. All P-values are unadjusted for multiple testing.

| | |
|---|---|

# Reporting Summary

Please do not complete any field with "not applicable" or n/a. Refer to the help text for what text to use if an item is not relevant to your study.
For final submission: please carefully check your responses for accuracy; you will not be able to make changes later.

## Statistics

For all statistical analyses, confirm that the following items are present in the figure legend, table legend, main text, or Methods section.

| n/a | Confirmed | |
|---|---|---|
| ☐ | ☒ | The exact sample size (*n*) for each experimental group/condition, given as a discrete number and unit of measurement |
| ☒ | ☐ | A statement on whether measurements were taken from distinct samples or whether the same sample was measured repeatedly |
| ☐ | ☒ | The statistical test(s) used AND whether they are one- or two-sided<br>*Only common tests should be described solely by name; describe more complex techniques in the Methods section.* |
| ☐ | ☒ | A description of all covariates tested |
| ☐ | ☒ | A description of any assumptions or corrections, such as tests of normality and adjustment for multiple comparisons |
| ☐ | ☒ | A full description of the statistical parameters including central tendency (e.g. means) or other basic estimates (e.g. regression coefficient) AND variation (e.g. standard deviation) or associated estimates of uncertainty (e.g. confidence intervals) |
| ☐ | ☒ | For null hypothesis testing, the test statistic (e.g. *F*, *t*, *r*) with confidence intervals, effect sizes, degrees of freedom and *P* value noted<br>*Give P values as exact values whenever suitable.* |
| ☐ | ☒ | For Bayesian analysis, information on the choice of priors and Markov chain Monte Carlo settings |
| ☒ | ☐ | For hierarchical and complex designs, identification of the appropriate level for tests and full reporting of outcomes |
| ☐ | ☒ | Estimates of effect sizes (e.g. Cohen's *d*, Pearson's *r*), indicating how they were calculated |

*Our web collection on statistics for biologists contains articles on many of the points above.*

## Software and code

Policy information about availability of computer code

| | |
|---|---|
| Data collection | Sequence data were demultiplexed, aggregated in BAM files and processed through a pipeline based on the Picard (v.2.1.1) suite of software tools and samtools (v.10). Reads were mapped onto human genome build 38 using BWA (v0.7.17). Variants were called using the Genome Analysis Toolkit (GATK; v4.1.4.1) HaplotypeCaller package to produce variant call set (VCF) files. Variants were further filtered using Variant Quality Score Recalibration (VQSR). Structural variants in UK Biobank were called using GraphTyper (2.7.1). |
| Data analysis | Quality Control filtering of vcf files was performed using vcftools v0.1.15, bcftools v1.9, picard v2.22.2 and plink v1.90b, as outlined in the methods. Variants were annotated using Ensembl Variant Effect Predictor v101 with assembly GRCh38. Structural variants were called using GraphTyper (). The code for each software is available at the website of each package. Data manipulation and analysis were performed using R-4.13 with packages clusterProfiler (4.2.2), data.table (1.14.2), dplyr (1.0.9), gtools (3.9.2.1), HGNChelper (0.8.9), msigdbr (7.5.1), tibble (3.1.7) and tidyr (1.2.0). Plots were created using R-4.13 using additional packages ggplot2 (3.3.6) and ggrepel (0.9.1). The code for each of the R packages can be found in their associated vignettes. |

For manuscripts utilizing custom algorithms or software that are central to the research but not yet described in published literature, software must be made available to editors and reviewers. We strongly encourage code deposition in a community repository (e.g. GitHub). See the Nature Portfolio guidelines for submitting code & software for further information.

## Data

Policy information about availability of data

All manuscripts must include a data availability statement. This statement should provide the following information, where applicable:

- Accession codes, unique identifiers, or web links for publicly available datasets
- A description of any restrictions on data availability
- For clinical datasets or third party data, please ensure that the statement adheres to our policy

> Data for the BRIDGES and PERSPECTIVE studies are available on reasonable request via the BCAC Data Access Co-ordinating Committee (BCAC@medschl.cam.ac.uk). Requests for UK Biobank should be made to the UK Biobank Access Management Team.

## Human research participants

Policy information about studies involving human research participants and Sex and Gender in Research.

| | |
|---|---|
| Reporting on sex and gender | Overall numbers of female and male subjects have been presented. Both males and females were included but, as breast cancer was the primary disease of interest, the number of male cases was insufficient to consider separate analyses by sex. Individuals whose genetically determined sex differed from their self-reported sex or gender were excluded. |
| Population characteristics | Participant numbers are summarised in Supplementary Tables 1 and 2. Diagnoses: invasive or in-situ breast cancer and control subjects without breast cancer. Genetic ancestry: individuals from MYBRCA and SGBCC studies were of east Asian ancestry, individuals in the remaining BCAC studies and UK Biobank studies were of European ancestry. Age distribution: UK biobank subjects were aged 38 - 73 years at recruitment (mean 56.8) and 14-82 at diagnosis (mean 57.1) BCAC cases were 18 - 86 year at diagnosis (mean 47.4), BCAC controls were 18 - 84 (mean 54.4) at last observation. Cases selected for this project we preferentially selected for early age at onset and/or family history of breast cancer. Known carriers of pathogenic germline BRCA1, BRCA2 and PALB2 variants, at the time of sample selection, were excluded. |
| Recruitment | The recruitment of cases and controls for BCAC studies varied by contributing study. Some studies recruited through hospital clinics while others recruited through population-based cancer registries. Recruitment strategies are summarised in Supplementary Table 24. UK Biobank subjects were recruited from 22 sites across the UK, through NHS patient registers. Cases cases were identified through linkage to the national cancer registration systems and through self-report. |
| Ethics oversight | The organisations providing ethical approval for the contributing studies are summarised in Supplementary Table 20. |

Note that full information on the approval of the study protocol must also be provided in the manuscript.

# Field-specific reporting

Please select the one below that is the best fit for your research. If you are not sure, read the appropriate sections before making your selection.

☒ Life sciences ☐ Behavioural & social sciences ☐ Ecological, evolutionary & environmental sciences

# Life sciences study design

All studies must disclose on these points even when the disclosure is negative.

| | |
|---|---|
| Sample size | A total of 26,368 female cases, 217,673 female controls, 94 male cases and 191,820 male controls were included in the analysis after quality control exclusions. The aim was to utlise the largest available dataset in order to maximise the power to detect associations, so no sample size calculation was relevant. |
| Data exclusions | Data exclusions are detailed in the methods. Variants were excluded on the basis of low sequencing depth, allelic balance, missingness and deviation from Hardy-Weinberg equilibrium. Samples with missing calls for >15% were excluded. For closely related individuals, only one individual was retained. For UK Biobank, individuals with excess relatives in the dataset were excluded. For UK Biobank individuals of non-European ancestry were excluded while for BCAC, individual not of European or east Asian ancestry were excluded. |
| Replication | These association analyses, used all the available data - the results from the BCAC and UK Biobank were combined in a meta-analysis to maximise power. While the novel associated reached a stringent level of signficance, replication in additional datasets will be required. |
| Randomization | Not applicable, this is an observational study. |
| Blinding | This is not an experimental study and did not require blinding. Processing of the sequence data was conducted independently of phenotypic data. |

# Reporting for specific materials, systems and methods

We require information from authors about some types of materials, experimental systems and methods used in many studies. Here, indicate whether each material, system or method listed is relevant to your study. If you are not sure if a list item applies to your research, read the appropriate section before selecting a response.

## Materials & experimental systems

| n/a | Involved in the study |
|-----|----------------------|
| ☒ ☐ | Antibodies |
| ☒ ☐ | Eukaryotic cell lines |
| ☒ ☐ | Palaeontology and archaeology |
| ☒ ☐ | Animals and other organisms |
| ☒ ☐ | Clinical data |
| ☒ ☐ | Dual use research of concern |

## Methods

| n/a | Involved in the study |
|-----|----------------------|
| ☒ ☐ | ChIP-seq |
| ☒ ☐ | Flow cytometry |
| ☒ ☐ | MRI-based neuroimaging |

