## [Peer Review File · Nature Genetics]

Peer Review Information

Manuscript Title: Exome sequencing identifies novel susceptibility genes and defines the contribution of coding variants to breast cancer risk.

Corresponding author name(s): Professor Douglas (F) Easton

Reviewer Comments & Decisions:

Decision Letter, initial version:

19th Aug 2022

Dear Professor Easton,

First, please allow me to apologise for the delay in returning this decision to you. Thank you for bearing with me.

Your Letter, "Exome sequencing identifies novel susceptibility genes and defines the contribution of coding variants to breast cancer risk." has now been seen by 3 referees. You will see from their comments copied below that while they find your work of considerable potential interest, they have raised quite substantial concerns that must be addressed. In light of these comments, we cannot accept the manuscript for publication, but would be very interested in considering a revised version that addresses these serious concerns.

We hope you will find the referees' comments useful as you decide how to proceed. If you wish to submit a substantially revised manuscript, please bear in mind that we will be reluctant to approach the referees again in the absence of major revisions.

To guide the scope of the revisions, the editors discuss the referee reports in detail within the team, including with the chief editor, with a view to identifying key priorities that should be addressed in revision and sometimes overruling referee requests that are deemed beyond the scope of the current study. In this case, we would like all the reviewer comments to be addressed experimentally where necessary or textually where appropriate. Please ensure that all technical concerns are addressed as a matter of priority.

If you choose to revise your manuscript taking into account these reviewer and editor comments, please highlight all changes in the manuscript text file. At this stage we will need you to upload a copy of the manuscript in MS Word .docx or similar editable format.

*2) If you have not done so already please begin to revise your manuscript so that it conforms to our Letter format instructions, available [here](http://www.nature.com/ng/authors/article_types/index.html). Refer also to any guidelines provided in this letter.

[redacted]

If you wish to submit a suitably revised manuscript we would hope to receive it within 6 months. If you cannot send it within this time, please let us know. We will be happy to consider your revision so long as nothing similar has been accepted for publication at Nature Genetics or published elsewhere. Should your manuscript be substantially delayed without notifying us in advance and your article is eventually published, the received date would be that of the revised, not the original, version.

Thank you for the opportunity to review your work.

Sincerely,

Safia Danovi
Editor
Nature Genetics

Referee expertise:

Referee #1: cancer genetic epidemiology

Referee #2: breast cancer germline genetics

Referee #3: cancer genetic epidemiology

Reviewers' Comments:

Reviewer #1:

Remarks to the Author:

A. Large exome sequencing study to identify and characterize genetic contributions to risk for breast cancer characterized associations for several known genetic factors and identified several new genetic factors that associate with breast cancer risk.

B. There are some very novel elements of the mathematical modeling of familial risk provided in supplemental material. The large size has allowed some potentially interesting observations to be identified but some the robust and highly significant protective effects of MRPL16 which replicates well are not discussed. I believe that is a weakness of the current reporting of results. Similarly, while the manuscript states that results for known breast cancer related variants have been presented, it overlooks TP53 and STK11, which are very well known genetic factors with mutations that give penetrances of > 80% for breast cancer in carriers in several series, therefore, I do not understand how these genes can be overlooked by these authors. Also, missing is any discussion of risk estimation or gene identification by age of the case/control pairs. Many of these genetic factors are most pronounced in increasing risk among younger women and the lack of any discussion of age as a contributing factor to gene identification is also a significant oversight.

C. Data and methodology. The approach to analysis across sources of data for the primary analyses seems generally sound except for oversights described above. However, I also wondered why the authors did not consider evaluating large deletions which should also be identifiable from exome sequencing data. In particular, it is very well known that about 10% of BRCA1 mutations are due to large deletions and these should be identifiable as true positives. One may also identify some additional genes by studying large deletions.

D. Generally well developed and well presented statistical plans and discussions. This element was highly innovative.

E. Conclusions. I am dubious about the extrapolation for the number of genes that could be identified given the limited explanation of risk that can be identified by the current analysis, particularly when the analyses are to my mind incomplete (no discussion of some known genes, lack of discussion of age, lack of study of deletions). Therefore, I think the space devoted to the discussion of missing genes is speculative and should be placed in supplement if it is going to be retained. I also think that some discussion of the protective factor MRPL16 would be of interest. I wondered if the discussion might also point to the risk estimates that are obtained from this unselected series.

F. As mentioned while what is presented is generally well written there are many components that are missing (age as a cofactor, genetic factors not studied, lack of development of study of protective factors). Also, while detailed analyses were conducted in supplement for selected genes by ER status and other cofactors, I do not know why these covariate analyses are not presented more systematically for all the genes that are highlighted as significant. The lack of a systematic approach to summarizing findings is frustrating to the reader.

G. References seemed ok.

H. Clarity. Generally well written. The supplementary mathematical development is very useful.

Reviewer #2:

Remarks to the Author:

The manuscript by Wilcox and colleagues describes results from exome sequencing of germline DNA from breast cancer cases and unaffected controls that was performed in order to identify new breast cancer susceptibility genes. The study uses a meta-analysis to combine data from several different sources including a BCAC study of family enriched breast cancer cases and controls that have been used widely for GWAS and were subjected to small gene panel sequencing (BRIDGES), a set of high and moderate risk family affected probands from PERSPECTIVE that were subjected to exome sequencing, and breast cancer cases and unaffected women from germline exome sequencing of the UK Biobank.

The authors identify known predisposition genes (validating their approach) and also identify pooled variants in ATRIP, MAP3K1 and SAMHD1 that appear to be associated with clinically relevant (odds ratio > 2) risk of breast cancer. Pooled protein truncating variants (PTVs) account for the significant associations with ATRIP and MAP3K1 whereas predicted deleterious missense were pooled with PTVs for the SAMHD1 associations. Importantly the associations were significant after adjustment for genome wide analyses.

The findings are potentially important because inherited variants in the three genes may allow identification of women at increased risk who could potentially benefit from enhanced screening and early detection of disease, although the numbers of women involved are likely small due to the rarity of the inactivating variants.

However, the methods used do raise some concerns. I'm not fully convinced by the evidence presented as there is variability between the results in the component studies. It is particularly odd when combined PTVs are associated with stronger risk of breast cancer in the population based UK biobank than in the family history enriched BRIDGES study. This suggests instability in the associations and effects of other genetic and non-genetic factors. While the ER based analyses are informative, other factors may be involved. I would like to see a further validation of the data in cohorts that account for the influence of family history, which on its own may confer ORs of 1.8 and may account for some of these observations. Furthermore, why do the authors think that 7 of 24 candidates significantly associated with risk confer protective effects. Doesn't this point to instability in the results. Perhaps the authors can comment on this issue in the text.

I also am concerned about the use of the CADD in silico prediction model to identify potentially deleterious missense variants, which account for the associations with SAMHD1. This method does not work well for missense variants in most of the known predisposition

genes so I'm not sure why it is shown here. Indeed BRIDGES has already published that Helix and other methods are better for known predisposition genes.

I did not see any mention of how the authors accounted for variants influencing splicing of these genes. Splicing defects account for many deleterious variants in several known predisposition genes. I would also like to see the authors account for the effects of nonsense mediated RNA decay by excluding the last 50bp of the penultimate exon in addition to the last exon of each MANE or Refseq transcript for each gene. Furthermore, the methods used do not account for the possibility of large genomic rearrangements in the candidate genes. For all of these reasons a number of key variants may be overlooked or inappropriately included.

Minor points

The authors should consider including case numbers in the analyses stratified by pathology

Reviewer #3:

Remarks to the Author:

Wilcox et al perform an exome wide association study for breast cancer in UK and Canadian population. They identified two new genes, ATRIP and MAP3K1, at which protein truncating variants are associated with breast cancer. They also identify one new gene, SAMHD1, at which deleterious missense variants are associated with breast cancer.

Overall the contribution of 3 new genes is important and the 3 genes are all highly plausible, so overall the results are very credible. In addition, the manuscript makes estimates about the contribution of rare coding sequence variation to the overall familial relative risk that may be important for future research about the genetics of breast cancer.

However, there are several issues with the manuscript which the authors should address:

1. The choice of analytical approaches in UK biobank vs. Bridges and Perspective is confusing and poorly justified. The UK biobank uses a method in which family history is weighted (genotype becomes the dependent variable in the logistic regression and disease + 1/2 family history is the predictor). It seems that this is an adaptation of a method previously described for GWAS (PMID: 28092683, which should be cited). Then the authors choose NOT to use this method for the analysis of Bridges and Perspective. This is not justified and it seems that it would actually be more helpful to weight family history in the Bridges and Perspective data where more of the cases would have a positive family history.

2. The meta-analysis method uses an approach to standardize the weights by the ratio of Z scores to the CHEK2 effect. The authors rationalize this choice by noting that the different studies use different selection criteria for family history and that the UKB was adjusted for family history (though see #1 - the same analytical approach could have been applied to both studies). This approach assumes heterogeneity of effects is uniform across genes. This is not necessarily the case as genes that affect ER-negative breast cancer risk likely have more of an effect in earlier life and genes that affect ER-positive breast cancer likely have an effect in later life. Thus, the heterogeneity of effects across different genes could be a function of age distributions as well as other ascertainment differences. In addition, standardizing the ratio to one gene seems inherently less robust than standardizing the effects to many known breast cancer susceptibility genes (e.g. BRCA1, BRCA2, PALB2, etc) which would have many more carriers and therefore produce a more stable estimate.

Finally, this reviewer is not aware of any literature to support standardizing weights for a meta-analysis based on effect sizes.

The generally accepted approach to deal with heterogeneous effects is to conduct a random effects meta-analysis rather than a fixed effects meta-analysis. Presumably the authors chose their approach as a way of avoiding random effects meta-analysis which tends to be underpowered. A thorough discussion of fixed effects and random effects models in genetic studies is given in PMID: 21565292 along with alternative tests for random effects that are more powerful.

3. The estimation of the familial risk contributed by different genes is an important analysis and suggests that the residual familial explained by coding sequence variation is very modest. However, this analysis is dependent on the assumptions that are made: namely that (a) the contribution of other genes is via deleterious missense variants and/or protein truncating (b) an autosomal dominant model. This model has been consistently true for the genes discovered to date. But there may be other genes that act in other ways. Therefore, the statements about residual familial risk due to coding variants should likely include a statement of limitation about model assumptions.

4. Data sharing: The authors state that data will be shared upon “reasonable request.” This is far below the minimum data sharing standards for modern papers. While it is assumed that sharing raw sequence data may be limited by genetic privacy concerns, summary statistics for the whole exome with the number of variants would not violate genetic privacy and would allow other investigators to use these results for meta-analysis, etc.

Author Rebuttal to Initial comments

NG-LE60351 Response to Reviewer’s Comments

We thank the reviewers for their helpful comments and have revised the paper accordingly. Our responses to the specific issues raised are as follows (responses in italics).

Reviewer #1:

Remarks to the Author:

A. Large exome sequencing study to identify and characterize genetic contributions to risk for breast cancer characterized associations for several known genetic factors and identified several new genetic factors that associate with breast cancer risk.

B. There are some very novel elements of the mathematical modeling of familial risk provided in supplemental material. The large size has allowed some potentially interesting observations to be identified but some the robust and highly significant protective effects of MRPL16 which replicates well are not discussed. I believe that is a weakness of the current reporting of results. Similarly, while the manuscript states that results for known breast cancer related variants have been presented, it overlooks TP53 and STK11, which are very well known genetic factors with mutations that give penetrances of > 80% for breast cancer in carriers in several series, therefore, I do not understand how these genes can be overlooked by these authors. Also, missing is any discussion of risk estimation or gene identification by age of the case/control pairs. Many of these genetic factors are most pronounced

in increasing risk among younger women and the lack of any discussion of age as a contributing factor to gene identification is also a significant oversight.

The revised analysis now includes a significantly larger dataset from UK Biobank. In the revised dataset, MRPL16 is no longer significant at $P < 0.001$ (indicating that the MRPL16 association seen originally was probably due to chance). Two genes (KRT28 and ADCY7) do show protective associations at this level, compared with 28 that are associated with increased risk. However, the associations could be due to chance (approximately 10 associations in each direction would be expected by chance). In view of this and space considerations, we did not discuss individually all the genes that reached this level of significance.

While TP53 and STK11 are established breast cancer susceptibility genes, variants in these genes are very rare and it is very difficult to evaluate their association in population-based studies. There was only 3 TP53 case-carriers and 1 STK11 PTV case-carrier in the datasets. We included results for TP53 in Supplementary Table 4 and have now updated these to include STK11.

We have now performed additional analysis restricting cases to age < 50. The results for this are in Supplementary Tables 5 and 6 and described in the text. No additional genes at exome-wide significance were found, though interestingly 40 genes showed positive associations at $P < 0.001$, suggesting that there is an enrichment of associations in this age-group.

C. Data and methodology. The approach to analysis across sources of data for the primary analyses seems generally sound except for oversights described above. However, I also wondered why the authors did not consider evaluating large deletions which should also be identifiable from exome sequencing data. In particular, it is very well known that about 10% of BRCA1 mutations are due to large deletions and these should be identifiable as true positives. One may also identify some additional genes by studying large deletions.

The primary analyses did not include large deletions or duplications since these are not straightforward to call reliably from exome sequence data and are not typically included in exome sequence analysis. Moreover, while it is true that for some genes such as BRCA1 there is an important contribution from large rearrangements, for most genes their frequency is small compared to small indels and SNVs and any gain in power for association would be very modest. We did however attempt to estimate the additional contribution that large rearrangements would make to the overall FRR.

To adjust the contribution of PTVs to reflect large rearrangements, we used whole genome sequence data that are available on a subset of the UK Biobank cohort to estimate the relative frequency of rearrangements. These were then used to adjust the carrier frequencies used in the familial relative risk calculations. Using this approach, the estimated FRR for all genes was 15.5%, compared with 13.9% if large rearrangements were ignored. This difference is largely explained by the contribution of BRCA1, where, as expected, the contribution of large rearrangements is significant (8.48% vs 7.41%).

D. Generally well developed and well presented statistical plans and discussions. This element was highly innovative.

We thank the reviewer for these comments.

E. Conclusions. I am dubious about the extrapolation for the number of genes that could be identified given the limited explanation of risk that can be identified by the current analysis, particularly when the analyses are to my mind incomplete (no discussion of some known genes, lack of discussion of age, lack of study of deletions). Therefore, I think the space devoted to the discussion of missing genes is speculative and should be placed in supplement if it is going to be retained. I also think that some discussion of the protective factor MRPL16 would be of interest. I wondered if the discussion might also point to the risk estimates that are obtained from this unselected series.

While accepting that the exact number of missing genes is speculative, we believe that the estimation of the overall contribution of PTVs is an important and novel contribution, as also noted by reviewer #3. In particular, the observation that only a small additional fraction of the familial risk is likely to be due to other coding variation is important in terms of directing future research.

As noted above, we have now included analyses by age and incorporated the effect of large rearrangements. We have not emphasised the risk estimates for individual genes since the risk estimates for new genes are generally unreliable in discovery studies, due to winner's curse, though the high relative risk for MAP3K1 is striking. We have commented on this in the discussion.

F. As mentioned while what is presented is generally well written there are many components that are missing (age as a cofactor, genetic factors not studied, lack of development of study of protective factors). Also, while detailed analyses were conducted in supplement for selected genes by ER status and other cofactors, I do not know why these covariate analyses are not presented more systematically for all the genes that are highlighted as significant. The lack of a systematic approach to summarizing findings is frustrating to the reader.

We have attempted to present the effects of age and pathological characteristics more systematically for all the genes highlighted as significant (with $P < 0.0001$ for the PTV meta-analysis or $P < 0.0001$ for PTV and deleterious rare missense variant meta-analysis). For completeness, we have also included the previously known genes. We do not think it would be appropriate to analyse the effect of other risk factors (genetic or lifestyle). The study lacks the power to examine these effects, even for the known genes.

In searching for novel associations, we focused on subtypes defined by ER status (also PR and TN disease) since this is the most important subdivision aetiologically and as noted, several of the known genes show effects that vary by ER status. As noted above, we also conducted exome-wide analyses for cases diagnosed at <50 years.

G. References seemed ok.

H. Clarity. Generally well written. The supplementary mathematical development is very useful.

We thank the reviewer for their comments.

Reviewer #2:

Remarks to the Author:

The manuscript by Wilcox and colleagues describes results from exome sequencing of germline DNA

from breast cancer cases and unaffected controls that was performed in order to identify new breast cancer susceptibility genes. The study uses a meta-analysis to combine data from several different sources including a BCAC study of family enriched breast cancer cases and controls that have been used widely for GWAS and were subjected to small gene panel sequencing (BRIDGES), a set of high and moderate risk family affected probands from PERSPECTIVE that were subjected to exome sequencing, and breast cancer cases and unaffected women from germline exome sequencing of the UK Biobank.

The authors identify known predisposition genes (validating their approach) and also identify pooled variants in ATRIP, MAP3K1 and SAMHD1 that appear to be associated with clinically relevant (odds ratio > 2) risk of breast cancer. Pooled protein truncating variants (PTVs) account for the significant associations with ATRIP and MAP3K1 whereas predicted deleterious missense were pooled with PTVs for the SAMHD1 associations. Importantly the associations were significant after adjustment for genome wide analyses.

The findings are potentially important because inherited variants in the three genes may allow identification of women at increased risk who could potentially benefit from enhanced screening and early detection of disease, although the numbers of women involved are likely small due to the rarity of the inactivating variants.

However, the methods used do raise some concerns. I'm not fully convinced by the evidence presented as there is variability between the results in the component studies. It is particularly odd when combined PTVs are associated with stronger risk of breast cancer in the population based UK biobank than in the family history enriched BRIDGES study. This suggests instability in the associations and effects of other genetic and non-genetic factors. While the ER based analyses are informative, other factors may be involved. I would like to see a further validation of the data in cohorts that account for the influence of family history, which on its own may confer ORs of 1.8 and may account for some of these observations. Furthermore, why do the authors think that 7 of 24 candidates significantly associated with risk confer protective effects. Doesn't this point to instability in the results. Perhaps the authors can comment on this issue in the text.

There are inevitably differences in the point estimates of the ORs between UK Biobank and the BCAC components, however, the confidence limits are wide and none of the effect sizes for the novel genes differ significantly. The estimates for BRCA1, BRCA2 and PALB2 are somewhat lower in the BCAC than in UK Biobank but this may have been influenced by the fact that known carriers in these genes were excluded from the BCAC exome sequencing. It is notable that for CHEK2 and ATM, the two most clearly established moderate-risk genes which did not influence the sample selection, the OR estimates are slightly higher in the BCAC, as expected.

It is true that differential selection for family history may influence the effect sizes, and hence the effect sizes in the BCAC and UK Biobank datasets may not be directly comparable. However, note that family history cannot generate a false positive association under the null hypothesis of no association, the association tests will always be valid even if the effect sizes are not directly comparable. The primary aim of these analyses, however, is discovery rather than effect size estimation, and for this, to only requirement is an effects are in the same direction and broadly comparable. We have expanded the discussion that effect sizes for the novel genes are in any case unreliable due to the winner's curse and

that future large replication will be required to obtain more reliable risk estimates and to examine the combined effect with other risk factors (which would include family history).

As noted in the response to reviewer #1, for only two genes (in the updated analyses) were PTVs associated with a reduced risk of breast cancer significant at $P < 0.001$, compared with 28 associated with an increased risk. The protective associations are therefore consistent with chance associations (whereas the large excess of positive associations indicates that many of these are likely to be real).

I also am concerned about the use of the CADD in silico prediction model to identify potentially deleterious missense variants, which account for the associations with SAMHD1. This method does not work well for missense variants in most of the known predisposition genes so I'm not sure why it is shown here. Indeed BRIDGES has already published that Helix and other methods are better for known predisposition genes.

As suggested by the reviewer, we have now updated the analysis to include an alternative definition of pathogenic variants based on Helix scores. We previously showed in the BRIDGES dataset that Helix scores performed better than other methods for several of the known susceptibility genes. This did not identify any additional associated genes at exome-wide significance, but some additional associations at $P < 0.0001$ were identified, including MDM4.

I did not see any mention of how the authors accounted for variants influencing splicing of these genes. Splicing defects account for many deleterious variants in several known predisposition genes. I would also like to see the authors account for the effects of nonsense mediated RNA decay by excluding the last 50bp of the penultimate exon in addition to the last exon of each MANE or Refseq transcript for each gene. Furthermore, the methods used do not account for the possibility of large genomic rearrangements in the candidate genes. For all of these reasons a number of key variants may be overlooked or inappropriately included.

As suggested by the reviewer, in the revised analysis we have now excluded PTVs in the last 50bp of the penultimate exon in addition to the last exon to avoid the inclusion of variants that do not lead to nonsense mediated RNA decay. The reviewer is certainly correct that aberrant splicing accounts for a proportion of pathogenic variants in the known genes. However, identifying the relevant non-canonical splice variants, even for the known genes, is a complex ongoing task involving both in-silico and in-vitro analyses. In this discovery analysis, we therefore took the usual pragmatic approach of including only canonical splice variants as likely deleterious PTVs. We have clarified this further in the methods. Further analysis of splicing for the newly identified susceptibility genes would be a subject for future research.

The reviewer is correct that we did not consider large rearrangements, and these may be responsible for additional susceptibility variants; unfortunately, these data are not currently available. Ignoring such variants may have reduced power somewhat (though even in the known genes only a minority of pathogenic variants are large rearrangements) but should not lead to bias. We have, however, adjusted our estimate of the contribution of PTVs in all genes to the FRR to account for large rearrangements – see also response to reviewer #1.

Minor points

The authors should consider including case numbers in the analyses stratified by pathology

These numbers have been included in Supplementary Tables 11 and 12.

Reviewer #3:

Remarks to the Author:

Wilcox et al perform an exome wide association study for breast cancer in UK and Canadian population. They identified two new genes, ATRIP and MAP3K1, at which protein truncating variants are associated with breast cancer. They also identify one new gene, SAMHD1, at which deleterious missense variants are associated with breast cancer.

Overall the contribution of 3 new genes is important and the 3 genes are all highly plausible, so overall the results are very credible. In addition, the manuscript makes estimates about the contribution of rare coding sequence variation to the overall familial relative risk that may be important for future research about the genetics of breast cancer.

However, there are several issues with the manuscript which the authors should address:

1. The choice of analytical approaches in UK biobank vs. Bridges and Perspective is confusing and poorly justified. The UK biobank uses a method in which family history is weighted (genotype becomes the dependent variable in the logistic regression and disease + 1/2 family history is the predictor). It seems that this is an adaptation of a method previously described for GWAS (PMID: 28092683, which should be cited). Then the authors choose NOT to use this method for the analysis of Bridges and Perspective. This is not justified and it seems that it would actually be more helpful to weight family history in the Bridges and Perspective data where more of the cases would have a positive family history.

We have now cited the Liu et al paper in the methods as requested. We note that our method differs from those suggested by Liu et al in that family history is included with a weight of ½ rather than 1, or using a 2-degree freedom test. As suggested by the reviewer, we have now redone the analysis also incorporating the family history data into the BCAC dataset. In fact, the approach is more informative for UK Biobank, because UK Biobank has a much larger number of controls for whom the family data can be utilised, including all the males. The gain for Bridges and Perspective is smaller and there is some disadvantage in that the exact conditional test for case-control data cannot now be used. However, we agree that using a uniform methodology across all the studies simplifies the presentation.

2. The meta-analysis method uses an approach to standardize the weights by the ratio of Z scores to the CHEK2 effect. The authors rationalize this choice by noting that the different studies use different selection criteria for family history and that the UKB was adjusted for family history (though see #1 - the same analytical approach could have been applied to both studies). This approach assumes heterogeneity of effects is uniform across genes. This is not necessarily the case as genes that affect ER-negative breast cancer risk likely have more of an effect in earlier life and genes that affect ER-positive breast cancer likely have an effect in later life. Thus, the heterogeneity of effects across different genes could be a function of age distributions as well as other ascertainment differences. In addition, standardizing the ratio to one gene seems inherently less robust than standardizing the effects to many known breast cancer susceptibility genes (e.g. BRCA1, BRCA2, PALB2, etc) which would have many more carriers and therefore produce a more stable estimate. Finally, this

reviewer is not aware of any literature to support standardizing weights for a meta-analysis based on effect sizes.

We note first that a weighted meta-analysis will give a valid test whatever weights are used – whilst standard (inverse variance weights) are commonly used, it is valid to use the different weights. We are not aware of this being used in genetic association studies but it is a logical approach and has been used, for example, in meta-analyses of clinical trials with different dosages.

We considered the possibility of using the effects of multiple genes to calibrate the Z-score weighting but rejected this for several reasons. The risks associated with CHEK2 PTVs have been reliably estimated and are in the “moderate” risk (2-4) range. Thus the test was calibrated to be most powerful for gene variants in this range. BRCA1, BRCA2 and PALB2 variants cause higher risks, but we deemed it was less likely that novel genes of this type would be identified (as they would have been more likely to be found already through family studies). The CHEK2 weight is the most reliably estimated; in addition, there is some arbitrariness in combining the estimates from multiple genes.

As a sensitivity analysis, we have also derived a meta-analysis weighting based on the five established genes. This gives relatively slightly more weight to UK Biobank (BCAC:UKB 0.307 vs 0.473 in our main analysis based on the CHEK2 estimates, however the genes reaching exome-wide significance (indeed the 10 most significant genes) are unchanged. We have noted this in the methods. Thus, and for the reasons given above, we believe that the CHEK2 weightings give a robust approach.

The generally accepted approach to deal with heterogeneous effects is to conduct a random effects meta-analysis rather than a fixed effects meta-analysis. Presumably the authors chose their approach as a way of avoiding random effects meta-analysis which tends to be underpowered. A thorough discussion of fixed effects and random effects models in genetic studies is given in PMID: 21565292 along with alternative tests for random effects that are more powerful.

As is common with most analyses of genetic association studies, we used a fixed effects approach since the effects are expected (a priori and allowing for the different sampling frames as discussed above) to be similar. The Han and Elskin paper noted by the reviewer is very interesting but its motivation is the potential differential effects due to LD in GWAS. This allows arbitrary differences in the effect sizes among populations (even in the opposite direction). This is not relevant here since the analyses are based on the burden of (presumed) causative deleterious variants. We note that, in this context, there has been little or no evidence of heterogeneity among populations (for example, the relative risks associated with pathogenic variants in the known genes show no evidence of variation among European, Asian and African populations). So it is a reasonable supposition that the same will be true for novel genes. The weighted analysis has a different rationale, that the effects in the different studies (specifically UK Biobank vs the BCAC studies) are likely to be bigger/smaller by a roughly constant ratio, due to the different sampling strategy, rather than any intrinsic heterogeneity. We postulated that this would be a more powerful approach in this context.

To further evaluate this, we also computed the Han Elskin statistics for all the genes achieving $P < 10^{-4}$ for PTVs. With the exception of BRCA1 (marginally so), BRCA2 and PALB2, the default fixed effect meta-analysis gave more significant P-values, and the heterogeneity parameter (τ) converged to zero. For these high-risk genes, there are large differences in the point estimates which overcome the loss of power

due to allowing a more general alternative. As noted in the response to reviewer #1, these genes are atypical in that the variants confer high risks and previously known carriers were excluded from the BCAC dataset. We conclude therefore that, for the genes we are predominantly aiming to identify, a weighted fixed-effect meta-analysis is a powerful approach and preferable. Indeed we also note that Han and Elskin note that “In the usual meta-analysis where one collects similar studies and expects the common effect of the variant, the results found by FE should be the top priority, but the results found by our method can also suggest interesting regions.”

We have summarised these additional sensitivity analyses in the methods.

3. The estimation of the familial risk contributed by different genes is an important analysis and suggests that the residual familial explained by coding sequence variation is very modest. However, this analysis is dependent on the assumptions that are made: namely that (a) the contribution of other genes is via deleterious missense variants and/or protein truncating (b) an autosomal dominant model. This model has been consistently true for the genes discovered to date. But there may be other genes that act in other ways. Therefore, the statements about residual familial risk due to coding variants should likely include a statement of limitation about model assumptions.

We have added a comment on the potential limitations on the estimate of the contribution to the familial risk. The analysis does assume dominant inheritance: however, a recessive component (due to rare coding variants) would contribute to the familial risk to siblings but not the familial risk to parent/offspring. So the results can therefore be interpreted as the contribution to the FRR to the offspring of affected individuals. The contribution of recessive genes is hard to estimate since homozygotes are very rare for most genes, but there is limited evidence of a higher familial risk of breast cancer to siblings that would indicate an important rare recessive component. We have also added a comment on this in the methods.

4. Data sharing: The authors state that data will be shared upon “reasonable request.” This is far below the minimum data sharing standards for modern papers. While it is assumed that sharing raw sequence data may be limited by genetic privacy concerns, summary statistics for the whole exome with the number of variants would not violate genetic privacy and would allow other investigators to use these results for meta-analysis, etc.

We completely agree with the reviewer that data sharing is very important and undertake to release summary statistics as requested. The summary statistics for BCAC will be made available through the BCAC website to coincide with publication. We have updated the Data and Code Availability Statement to cover this. While we cannot ourselves release data for UK Biobank, we will return summary statistics for UK Biobank so they (and indeed the raw data) will be available through the UK Biobank Access Management Team.

Decision Letter, first revision:

17th Feb 2023

Dear Professor Easton,

First, please accept my apologies for the delay in returning this decision to you. Thank you for bearing with me.

Your Letter, "Exome sequencing identifies novel susceptibility genes and defines the contribution of coding variants to breast cancer risk." has now been seen by 2 referees. Unfortunately, Reviewer #2 did not submit a report so we asked Reviewer #1 to look over your response to their comments. These comments have been included in an attachment.

You will see from their comments below that while they find your revisions to have improved the paper, some remaining points have been raised. We are interested in the possibility of publishing your study in Nature Genetics, but would like to consider your response to all these concerns in the form of a revised manuscript before we make a final decision on publication.

We therefore invite you to revise your manuscript taking into account all reviewer and editor comments. Please highlight all changes in the manuscript text file. At this stage we will need you to upload a copy of the manuscript in MS Word .docx or similar editable format.

*2) If you have not done so already please begin to revise your manuscript so that it conforms to our Letter format instructions, available [here](http://www.nature.com/ng/authors/article_types/index.html). Refer also to any guidelines provided in this letter.

[REDACTED]

We hope to receive your revised manuscript within four to eight weeks. If you cannot send it within this time, please let us know.

Sincerely,

Safia Danovi
Editor
Nature Genetics

Reviewers' Comments:

Reviewer #1:

Remarks to the Author:

The reviewers have adequately addressed all of my questions.

[Ed; in confidential comments to the editors, Reviewer #1 asked whether inserting a gene-set enrichment analysis for the genes that are significant at $p < 10^{-4}$ would help structure the potential novel findings and identify pathways involved in breast cancer risk. They were also unclear as to why certain certain genes were included in supplementary table 6. They cited WBP11 as an example as it is not significant overall or in the substrata. Please address these comments in your revision]

Reviewer #2:

[Ed; please refer to the attached document which details Reviewer #1's feedback regarding your response to Reviewer #2]

Reviewer #3:

Remarks to the Author:

The authors have made several improvements to the manuscript.

However, the justification for the weighting approach used in the meta-analysis. First, since the inverse variance weights are so commonly used in meta-analysis, the use of a different weighting scheme should, at least, be supported by a citation. Can the authors cite examples of meta-analyses, even if not in the genetics literature, that used different weighting schemes?

Second, the use of one gene to weight the studies still seems like a choice that is much less stable and robust than the use of multiple genes. It is reassuring that weighting approach using multiple genes did not change the results, but it would be better to actually present those results.

[Ed; we appreciate that there is some disagreement between reviewers about the weighting approach, but we think that following Reviewer #3's advice (which only require textual changes) would help strengthen confidence in the approach]

[Sent to Safia Danovi from Reviewer #1 following a request for them to look at the authors' response to Reviewer #2]

The authors responded to all of my questions and they also responded to almost all the comments from the other two reviewers. I disagree with reviewer 3 on several issues, most notably the statement that a random effects model should be used in this context, and I found the response by the authors correct. Also the comments from the authors about weighting are also correct and the reviewer is not correct. Therefore, other than the very short revision I suggest below, I think this paper is ready to be published. Basically they just need to give a little more information about the sample characteristics and what proportion of individuals did not have BRCA1/ BRCA2 or PALB2 sequenced and of these how many were affected versus not affected. Excluding affected individuals from the analysis will cause a bias towards the null which should be clarified a little in the text (it is stated but without details).

“It is particularly odd when combined PTVs are associated with stronger risk of breast cancer in the population based UK biobank than in the family history enriched BRIDGES study. This suggests instability in the associations and effects of other genetic and non-genetic factors. While the ER based analyses are informative, other factors may be involved. I would like to see a further validation of the data in cohorts that account for the influence of family history, which on its own may confer ORs of 1.8 and may account for some of these observations. Furthermore, why do the authors think that 7 of 24 candidates significantly associated with risk confer protective effects. Doesn't this point to instability in the results. Perhaps the authors can comment on this issue in the text. There are inevitably differences in the point estimates of the ORs between UK Biobank and the BCAC components, however, the confidence limits are wide and none of the effect sizes for the novel genes differ significantly. The estimates for BRCA1, BRCA2 and PALB2 are somewhat lower in the BCAC than in UK Biobank but this may have been influenced by the fact that known carriers in these genes were excluded from the BCAC exome sequencing. It is notable that for CHEK2 and ATM, the two most clearly established moderate-risk genes which did not influence the sample selection, the OR estimates are slightly higher in the BCAC, as expected.”

The exclusion of BRCA1, BRCA2 and PALB2 carriers from exome sequencing is noted in the manuscript but the authors need to provide more detail. How many carriers were excluded from this analysis? Also can they give a little more detail about the affection status of the

excluded individuals. I expect that does drive down effect estimates but without details it is impossible to know how much of an effect this exclusion might have had.

Author Rebuttal, first revision:

NG-LE60351R Response to Reviewer's Comments

We thank the reviewers for their additional comments and have revised the paper accordingly. Our responses to the issues raised are as follows (responses in italics).

Reviewer #1

[Ed; in confidential comments to the editors, Reviewer #1 asked whether inserting a gene-set enrichment analysis for the genes that are significant at $p < 10^{-4}$ would help structure the potential novel findings and identify pathways involved in breast cancer risk. They were also unclear as to why certain genes were included in supplementary table 6. They cited WBP11 as an example as it is not significant overall or in the substrata. Please address these comments in your revision]

As suggested, we have now completed gene-set enrichment analyses (Supplementary Tables 13-14 and text added). These analyses further emphasise the central role of DNA repair, as almost all the significant pathways were related to this. In addition, we performed additional heritability analyses based on tumour suppressor genes and breast cancer driver genes, as well as the lead pathways from the GSEA (Supplementary Table 16). These analyses show that the high proportion of genes in these categories are predicted to be risk genes.

With regard to Supplementary Table 6, this includes, for completeness, all genes significant at $P < 0.001$ for any of the subtypes (by ER, PR or TN status). This is indicated in the Table description. WBP11 is significant non-TN disease, hence its inclusion.

Reviewer #2 (response from reviewer #1)

The exclusion of BRCA1, BRCA2 and PALB2 carriers from exome sequencing is noted in the manuscript but the authors need to provide more detail. How many carriers were excluded from this analysis? Also can they give a little more detail about the affection status of the excluded individuals. I expect that does drive down effect estimates but without details it is impossible to know how much of an effect this exclusion might have had.

As requested, we have added details in the methods on the number of cases and controls BRCA1, BRCA2 and PALB2 carriers that were not considered in the exome sequencing. We should like to clarify that the individuals were not excluded from the analysis, they were not considered for selection for exome sequencing. It is not clear whether the exclusion of previously known carriers for would bias the risk estimates for these genes, however the main emphasis of the paper is in any case on the novel genes – since reliable estimates for BRCA1, BRCA2 and PALB2 have already been obtained in previous targeted sequencing studies.

Reviewer #3

However, the justification for the weighting approach used in the meta-analysis. First, since the inverse variance weights are so commonly used in meta-analysis, the use of a different weighting scheme should, at least, be supported by a citation. Can the authors cite examples of meta-analyses, even if not in the genetics literature, that

used different weighting schemes?

There is literature on meta-analyses in epidemiology and clinical trials which different studies have different levels of exposure and the interest is in the effect size per unit exposure ("doses"). The method here is equivalent but dose is represented by the $\log(OR)$ for the CHEK2 (or all known genes). We have elaborated on this in the methods section and included a reference to the Greenland and Longnecker paper which is a classic reference (though our analysis is a much simpler case in that there are effectively two studies and only one OR for each) and the Longnecker et al. paper which is an example they draw on.

Second, the use of one gene to weight the studies still seems like a choice that is much less stable and robust than the use of multiple genes. It is reassuring that weighting approach using multiple genes did not change the results, but it would be better to actually present those results.

[Ed; we appreciate that there is some disagreement between reviewers about the weighting approach, but we think that following Reviewer #3's advice (which only require textual changes) would help strengthen confidence in the approach]

As requested, we have also included results using the alternative weighting approach. These are given in Supplementary Table 17.

Decision Letter, second revision:

17th Mar 2023

Dear Dr. Easton,

Thank you for submitting your revised manuscript "Exome sequencing identifies novel susceptibility genes and defines the contribution of coding variants to breast cancer risk." (NG-LE60351R1). Your revisions were assessed in-house by the editorial team and I'm pleased to say that we'll be happy in principle to publish it in Nature Genetics, pending minor revisions to satisfy

Sincerely,

Safia Danovi
Editor
Nature Genetics

Final Decision Letter:

5th Jul 2023

Dear Dr. Easton,

I am delighted to say that your manuscript "Exome sequencing identifies novel susceptibility genes and defines the contribution of coding variants to breast cancer risk." has been accepted for publication in an upcoming issue of Nature Genetics.

Your paper will be published online after we receive your corrections and will appear in print in the next available issue. You can find out your date of online publication by contacting the Nature Press Office (press@nature.com) after sending your e-proof corrections. Now is the time to inform your Public Relations or Press Office about your paper, as they might be interested in promoting its publication. This will allow them time to prepare an accurate and satisfactory press release. Include your manuscript tracking number (NG-LE60351R2) and the name of the journal, which they will need when they contact our Press Office.

Please note that *Nature Genetics* is a Transformative Journal (TJ). Authors may publish their research with us through the traditional subscription access route or make their paper immediately open access through payment of an article-processing charge (APC). Authors will not be required to make a final decision about access to their article until it has been accepted. [Find out more about Transformative Journals](https://www.springernature.com/gp/open-research/transformative-journals)

Authors may need to take specific actions to achieve [compliance with funder and institutional open access mandates](https://www.springernature.com/gp/open-research/funding/policy-compliance-faqs). If your research is supported by a funder that requires immediate open access (e.g. according to [Plan S principles](https://www.springernature.com/gp/open-research/plan-s-compliance)) then you should select the gold OA route, and we will direct you to the compliant route where possible. For authors selecting the subscription publication route, the journal's standard licensing terms will need to be accepted, including [self-archiving and license to publish](https://www.nature.com/nature-portfolio/editorial-policies/self-archiving-and-license-to-publish). Those licensing terms will supersede any other terms that the author or any third party may assert apply to any version of the manuscript.

Please note that Nature Portfolio offers an immediate open access option only for papers that were first submitted after 1 January, 2021.

If you have not already done so, we invite you to upload the step-by-step protocols used in this manuscript to the Protocols Exchange, part of our on-line web resource, natureprotocols.com. If you complete the upload by the time you receive your manuscript proofs, we can insert links in your article that lead directly to the protocol details. Your protocol will be made freely available upon publication of your paper. By participating in natureprotocols.com, you are enabling researchers to more readily reproduce or adapt the methodology you use. [Natureprotocols.com](https://natureprotocols.com) is fully searchable, providing your protocols and paper with increased utility and visibility. Please submit your protocol to <https://protocolexchange.researchsquare.com/>. After entering your nature.com username and password you will need to enter your manuscript number (NG-LE60351R2). Further information can be found at <https://www.nature.com/nature-portfolio/editorial-policies/reporting-standards#protocols>

Sincerely,

Safia Danovi
Editor
Nature Genetics